# Parameter-free HE-friendly Logistic Regression

**Junyoung Byun**[*]
Seoul National University
Seoul, Korea
quswns95@snu.ac.kr

**Woojin Lee**[*]
Dongguk University-Seoul,
Seoul, Korea
wj926@dgu.ac.kr

**Jaewook Lee**[†]
Seoul National University
Seoul, Korea
jaewook@snu.ac.kr

## Abstract

Privacy in machine learning has been widely recognized as an essential ethical and legal issue, because the data used for machine learning may contain sensitive information. Homomorphic encryption has recently attracted attention as a key solution to preserve privacy in machine learning applications. However, current approaches on the training of encrypted machine learning have relied heavily on hyperparameter selection, which should be avoided owing to the extreme difficulty of conducting validation on encrypted data. In this study, we propose an effective privacy-preserving logistic regression method that is free from the approximation of the sigmoid function and hyperparameter selection. In our framework, a logistic regression model can be transformed into the corresponding ridge regression for the logit function. We provide a theoretical background for our framework by suggesting a new generalization error bound on the encrypted data. Experiments on various real-world data show that our framework achieves better classification results while reducing latency by $\sim 68\%$, compared to the previous models.

## 1 Introduction

Machine learning on encrypted data (MLED) is an effective method to ensure privacy for both machine learning models and data used for the model training. Unlike other privacy-preserving machine learning methods, MLED does not require decryption of intermediate products of the algorithm or data transfer between participants, enabling a complete outsourcing of machine learning. As data have become increasingly valuable, MLED models have been actively studied in an attempt to combine data and computing power that are separated from each other. With the development of efficient homomorphic encryption (HE) schemes that enable MLED, recent studies [4, 9, 38, 30, 29, 31] cover most machine learning models, from simple linear regression to deep neural networks (DNN).

However, current research on MLEDs remains lacking owing to the limitations of the proposed HE schemes. Operations on ciphertexts are very inefficient in terms of computation time and memory consumption, compared to operations on plaintexts. In addition, after performing a certain number of operations, a costly procedure called bootstrapping is required [19, 5]. Therefore, the majority of studies are limited to the inference phase, assuming a situation that provides machine learning inference as a service. The training phase should be studied more deeply top extract the maximum value from the data.

One of the MLED models, whose training steps have been commonly studied, is logistic regression (LR). LR is a simple classification model that linearly divides the data space, which is used effectively in various fields, including medicine, marketing, and geology [8, 10, 28]. [23] trained an encrypted LR model with gradient descent and approximated the sigmoid function with the most similar polynomials within a certain range. Despite performing well on several benchmark datasets, their

---

[*]Equal contribution.
[†]Corresponding author.

35th Conference on Neural Information Processing Systems (NeurIPS 2021).

methodology raises some concerns because the training parameters, such as learning rate, number of iterations, and range of sigmoid function approximation should be predetermined. Indeed, this problem is common for all proposed MLED training models.

In this study, we propose an effective privacy-preserving LR method that is free from most hyper-parameter selections. First, we train an unencrypted classification model to extract the prediction probability for each label. Then, we solve for a ridge regression that predicts the logit result of the estimated probability. Our theoretical results provide a new generalization error bound, which can be optimized by properly estimating the logit and our proposed mean matching, which aims to reduce the gap between the distributions of encrypted and unencrypted data. To benefit from the trade-off between security and efficiency, we define private variables, whose privacy is more important than the privacy of others, which has been widely recognized in other fields but has rarely been introduced into MLED. The experimental results show that, compared to the prior encrypted LR models, our method achieves better classification results and lower training latency.

## 2 Preliminary Information

Throughout this paper, vectors are written in bold lowercase and matrices in bold uppercase. A ciphertext is indicated in the form of the corresponding message enclosed in brackets (e.g. $[m]$). Operations between ciphertexts or between ciphertext and plaintext are both represented same as those between plaintexts.

**Fully Homomorphic Encryption** A fully HE (FHE) scheme consists of four procedures : KGen, Enc, Dec and Eval. KGen generates secret key $sk$ and public key $pk$ [18]. A message $m$ is encrypted to a ciphertext $[m] = \text{Enc}(m, pk)$, and a ciphertext $[m']$ is decrypted back to a plaintext $m' = \text{Dec}([m'], sk)$. Eval evaluates a function $f$ with a vector of ciphertexts $([m_1], \ldots, [m_k])$ as an input; thus, it holds that $\text{Dec}(\text{Eval}(([m_1], \ldots, [m_k]), f, pk), sk) = f(m_1, \ldots, m_k)$. An FHE scheme naturally enables addition and multiplication; therefore, non-polynomial operations must be approximated by polynomial operations. Noise is introduced when a message is first encrypted, which grows as the operation between ciphertexts proceeds. If the noise exceeds a certain level, the ciphertext cannot be decrypted correctly. FHE allows an unlimited number of operations through bootstrapping which resets the noise, but bootstrapping incurs a large computational cost. Therefore, leveled HE (LHE) which avoids bootstrapping is preferred in many studies, with the number of operations required and corresponding scheme parameters determined in advance. In the case of using gradient descent in MLED training, LHE is not suitable because the number of iterations is very limited; whereas our method is suitable for LHE because the number of required operations is not too large and can be determined before implementation.

**Logistic Regression with HE** The loss function of LR is the negative log-likelihood, which is given by

$$J(\boldsymbol{\theta}) = \sum_{i=1}^{n} \log(1 + \exp(-y^i \boldsymbol{\theta}^T \boldsymbol{x}^i)) \tag{1}$$

where $\boldsymbol{\theta}$ is the weight vector, $\boldsymbol{x}^i$ is i-th input and $y^i \in \{+1, -1\}$ is the label of i-th input. [22] minimized (1) with Nesterov's accelerated gradient, which is a slight modification of the gradient descent algorithm. The gradient descent algorithm can be written as follows:

$$\boldsymbol{\theta}_{t+1} = \boldsymbol{\theta}_t - \eta \boldsymbol{g}_t, \quad \boldsymbol{g}_t = \frac{\partial J}{\partial \boldsymbol{\theta}} = -\sum_{i=1}^{n} \text{sig}(-y^i \boldsymbol{\theta}_t^T \boldsymbol{x}^i) \cdot y^i \boldsymbol{x}^i \tag{2}$$

where $\eta$ is the learning rate and $\text{sig}(x) = 1/(1 + \exp(-x))$ is the sigmoid function. When implementing this model with HE, there are two major limitations. The first is the selection of the learning rate and number of iterations for the gradient descent. In Figure 1(a), we plotted how the loss changes according to the number of iterations for different learning rates. Observe that, the performance can vary drastically depending on the learning rate. In addition, the optimal learning rate is different depending on the number of iterations (compare the orange line ($\eta = 0.1$) and pink line ($\eta = 0.01$)). When dealing with plaintext, the optimal parameters can be found through cross validation, but this is impossible in MLED because the the weight vector cannot be decrypted during training. On the other hand, our method can obtain a closed-form solution without using gradient descent.

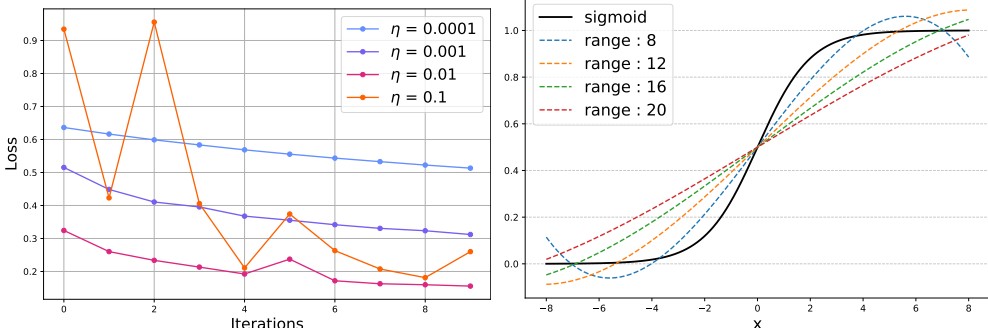

(a) Importance of choosing the appropriate parame-
ters in learning LR

(b) Approximation performance of sigmoid function
decreases as possible range of input grows

Figure 1: Sensitivity analysis on the hyperparameters in previous privacy preserving LR

Another limitation is that the sigmoid function cannot be evaluated efficiently using HE. Because the low-order Taylor expansion results in a good approximation only in a localized domain, [22] approximated the sigmoid function with a polynomial showing the lowest mean squared error in the domain $[-8, 8]$. Figure 1(b) shows the different approximation results according to the domain range when approximated by a 3rd order polynomial. The maximum error between the polynomial and the sigmoid function is $0.11$ when the range is $[-8, 8]$ (blue line), which increases to $0.26$ when the range is $[-20, 20]$ (red line). Therefore, it is essential to obtain a tight bound for the input value of the sigmoid function, which is not possible for encrypted data. Conversely, our method solves ridge regression on the ciphertext; thus, there is no need to evaluate the sigmoid function.

**Partial Encryption of Private Variables**   It has been widely accepted that privacy for some variables is more important than security for other variables. In the field of privacy-preserving data mining, k-anonymity [40] transforms the published variables to prevent exposure of personal identity. Subsequent concepts, such as l-diversity [32, 41] and t-closeness [27] were proposed to protect other private variables, including personal income and medical records. In fair machine learning studies [25, 42], which have gained popularity recently, variables that may cause discrimination are defined as sensitive variables, which are required to be not disclosed [20, 21]. In cryptography research, [39, 13] presented security criteria and variables were selectively encrypted accordingly.

When partially encrypting some private variables, the main purpose is to achieve a trade-off between security and computation & storage efficiency. In Chapter 3, we propose an efficient ridge regression method with a partially encrypted dataset that cannot be directly applied to LR. We separate operations on encrypted and non-encrypted variables, whereas in LR, the input of the sigmoid function is the weighted sum of all variables. In addition, simply applying the existing LR algorithm to partially encrypted data does not result in sufficient efficiency nor does it solve the problems of parameter selection. We solve this problem by replacing the LR with ridge regression for the pre-calculated logit.

## 3   Framework

In this study, we propose a new framework that can effectively handle classification tasks, while preserving the privacy of private variables. We set two different datasets for the same variables, $\mathcal{D}_1 = \{(\boldsymbol{x}_1^i, y_1^i) | i = 1, \ldots, n_1\}$ and $\mathcal{D}_2 = \{(\boldsymbol{x}_2^j, y_2^j) | j = 1, \ldots, n_2\}$, where $\boldsymbol{x}_2^j = (x_{21}^j, \ldots, x_{2p}^j, [x_{2(p+1)}^j], \ldots, [x_{2(p+\ell)}^j])$ contains $\ell$ encrypted private variables.

**Threat Model**   The participants of our framework consist of data owners O, a modeler M and a crypto-service provider C. We assume that the participants are honest-but-curious and do not collude, which is widely accepted in MLED studies [37, 34, 1, 35]. Our security goals are as follows:

- Neither M nor C should obtain information on private variables.

- Neither O nor C should obtain information on the learned model.

**Protocol**   The details of our protocol to achieve the security goal are as follows:

- (**Teacher modeling**) M trains a teacher model $f_s$ with an unencrypted dataset $\mathcal{D}_1$, where $\mathcal{D}_1$ is owned by M or publicly available.
- (**Encryption**) C generates keys $(pk, sk)$ and sends $pk$ to O and M. O encrypts the private variables of their dataset $\mathcal{D}_2$ and sends $\mathcal{D}_2$ to M.
- (**Training on encrypted data**) M infers encrypted logit $\mathsf{Enc}(l_2) = f_s(\mathcal{D}_2)$ and evaluates $\mathsf{Enc}(\tilde{l_2}) = \mathsf{Enc}(l_2) + \mathsf{Enc}(\beta)$ through mean matching. M trains the privacy-preserving ridge regression on $\mathcal{D}_2$ and $\mathsf{Enc}(\tilde{l_2})$ and obtains $\mathsf{Enc}(\omega)$.
- (**Decryption**) M generates a random polynomial $r$ and sends $\mathsf{Enc}(\omega + r)$ to C. C decrypts $w + r$, adds a random discrete Gaussian noise $e$, and sends $w + r + e$ back to M. M subtracts $r$, and the final weight is obtained as $w + e$.

Similar to [12, 16], the security of our protocol follows directly from the semantic security (against passive adversaries) of the underlying HE scheme. In our protocol, $e$ is added to defend against attack proposed by [26], which, according to [6], with a high probability causes only <1 bits of precision loss.

Our framework consists of three steps, excluding encryption and decryption. The first step is **teacher modeling**, which is the first of our protocol, and the second and third steps are **mean matching** and **ridge regression**, which corresponds with the third protocol.

**Teacher modeling**   In the first step, M trains a classification model with $\mathcal{D}_1$, which mimics the first phase of the knowledge distillation in the first model; then, we extract the soft probability from the target label. However, our method has several major differences from knowledge distillation. First, knowledge distillation aims to train a rather simple model that performs comparably to a complex teacher model, in which extracting the probability is a means to achieve the goal. On the other hand, in our framework the probability plays a more important role. We transform the classification task into a regression problem through the probability. Another difference is that in our method, the teacher model does not need to be a complex model. There are several advantages to using a simple model as a teacher model, which we will demonstrate later.

Meanwhile, we should assume that M should possess an unencrypted $\mathcal{D}_1$ that can be used in step 1. Considering that M is depicted as a company seeking to profit from their model in many studies [30, 29, 31], there may be individuals who provide their private information to M in return. To be more realistic, we assumed there are relatively few people who publish their private information. Moreover, the underlying distributions of $\mathcal{D}_1$ and $\mathcal{D}_2$ are heterogeneous because they have different features than those who do not want to provide private information. To fit the former setting, we use a simple model that generalizes sufficiently with a small amount of data as the teacher model.

**Mean matching**   In the second step, using the teacher model M infers the logit of $\mathcal{D}_2$. In the case where the teacher model uses LR, the inference is simply an inner product between the model parameter $\boldsymbol{\theta}$ and $\boldsymbol{x}^j$s, which can be efficiently computed with HE using slot-wise rotation (See the Allsum algorithm in [23]). Denoting the teacher model as $f_s$ and logits of $\mathcal{D}_1$ and $\mathcal{D}_2$ as $\boldsymbol{l}_1$ and $\boldsymbol{l}_2$, respectively, we can obtain $\boldsymbol{l}_2$ from $\mathcal{D}_2$ using the teacher model $f_s$. However, because $\mathcal{D}_1$ and $\mathcal{D}_2$ have different distributions, also $\boldsymbol{l}_1$ and $\boldsymbol{l}_2$ might. Therefore, rather than directly using the logit distribution $\boldsymbol{l}_2$, we suggest adding a regularization term $\beta$ to $\boldsymbol{l}_2$ so that it can consider the difference between two distributions. The $\beta$ should satisfy:

$$\frac{\sum_i f_s(x_1^i)}{\sum_i y_1^i} = \frac{\sum_i (f_s(x_2^i) + \beta)}{\sum_i y_2^i} \tag{3}$$

Now applying the regularization term $\beta$, we can obtain the shifted logit $\tilde{\boldsymbol{l}}_2$, where $\tilde{l}_2^i = f_s(x_2^i) + \beta$.

Note that (3) can be seen as a simplified version of kernel mean matching reported in [17], which is widely used in domain adaptation studies [24]. Kernel mean matching attempts to match the means of the distributions of $\boldsymbol{x}$s in a kernel space. When directly applying mean matching to logits, the distributions of logits are forced to become similar, which has a negative effect on moving the

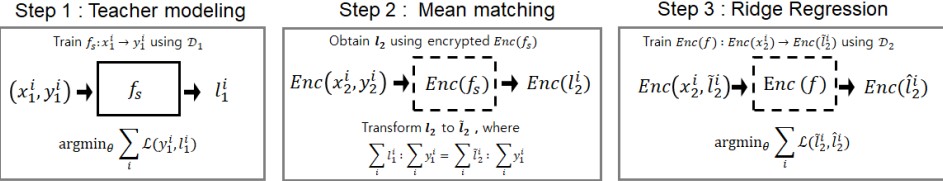

Figure 2: Overall framework of our proposed method

prediction away from the true label. Therefore, we multiplied the logit by a weight, which reflects the distribution of the true label.

**Ridge regression** Finally, we train a ridge regression on $\mathcal{D}_2$, where the target variable is the shifted logit $\tilde{l}_2$, not $\boldsymbol{y}_2$. Intuitively, using a well-estimated probability enables a training at least as accurate as that achieved when using a binary target; our theoretical results reported in Chapter 4 support this claim. By encrypting a small private portion of the entire information, we can train an efficient ridge regression without having to explore the learning parameters. Let $\boldsymbol{X} = ((\boldsymbol{x}_2^1)^T \cdots (\boldsymbol{x}_2^{n_2})^T)^T \in \mathbb{R}^{n_2 \times (p+\ell)}$, $[\boldsymbol{H}_s] = ([\boldsymbol{h}_1] \cdots [\boldsymbol{h}_\ell])$ be the encrypted $\ell$ columns of $\boldsymbol{X}$ and $\boldsymbol{X}_{(-s)}$ the other $p$ columns. Then the ridge estimate is

$$\hat{\boldsymbol{f}} = \boldsymbol{X}(\boldsymbol{X}^T\boldsymbol{X} + \lambda\boldsymbol{I}_{p+\ell})^{-1}\boldsymbol{X}^T[\tilde{l}_2] = \boldsymbol{X}\boldsymbol{X}^T(\boldsymbol{X}\boldsymbol{X}^T + \lambda\boldsymbol{I}_{n_2})^{-1}[\tilde{l}_2] = [\tilde{l}_2] - \lambda(\boldsymbol{X}\boldsymbol{X}^T + \lambda\boldsymbol{I}_{n_2})^{-1}[\tilde{l}_2]. \tag{4}$$

Because $\boldsymbol{X}\boldsymbol{X}^T = \boldsymbol{X}_{(-s)}\boldsymbol{X}_{(-s)}^T + [\boldsymbol{H}_s][\boldsymbol{H}_s^T]$, by applying the Sherman-Woodbury inversion formula we have

$$(\boldsymbol{X}\boldsymbol{X}^T + \lambda\boldsymbol{I}_{n_2})^{-1} = \boldsymbol{A}^{-1} - \boldsymbol{A}^{-1}[\boldsymbol{H}_s](\boldsymbol{I}_\ell + [\boldsymbol{H}_s^T]\boldsymbol{A}^{-1}[\boldsymbol{H}_s])^{-1}[\boldsymbol{H}_s^T]\boldsymbol{A}^{-1}$$

where $\boldsymbol{A} = \boldsymbol{X}_{(-s)}\boldsymbol{X}_{(-s)}^T + \lambda\boldsymbol{I}_{n_2}$. Using the singular value decomposition (SVD) of $\boldsymbol{X}_{(-s)} = \boldsymbol{U}\boldsymbol{\Sigma}\boldsymbol{V}^T$ where $\boldsymbol{U} \in \mathbb{R}^{n_2 \times n_2}$ and $\boldsymbol{V} \in \mathbb{R}^{p \times p}$ are orthogonal matrices, and $\boldsymbol{\Sigma} \in \mathbb{R}^{n_2 \times p}$ is a diagonal matrix with diagonal entries $\sigma_1 \geq \cdots \geq \sigma_p \geq \sigma_{p+1} = \cdots = \sigma_{n_2} = 0$, we have

$$\boldsymbol{A}^{-1} = (\boldsymbol{U}\boldsymbol{\Sigma}\boldsymbol{\Sigma}^T\boldsymbol{U}^T + \lambda\boldsymbol{I}_{n_2})^{-1} = \boldsymbol{U}(\boldsymbol{\Sigma}\boldsymbol{\Sigma}^T + \lambda\boldsymbol{I}_{n_2})^{-1}\boldsymbol{U}^T.$$

Then the ridge estimate can be rewritten as

$$\hat{\boldsymbol{f}} = [\tilde{l}_2] - \lambda\boldsymbol{A}^{-1}[\tilde{l}_2] + \lambda\left(\sum_{j=1}^{n_2}\frac{\boldsymbol{u}_j\boldsymbol{u}_j^T[\boldsymbol{H}_s]}{\sigma_j^2 + \lambda}\right)(\boldsymbol{I}_\ell + [\boldsymbol{\Xi}])^{-1}\left(\sum_{j=1}^{n_2}\frac{1}{\sigma_j^2 + \lambda}[\boldsymbol{H}_s^T]\boldsymbol{u}_j\boldsymbol{u}_j^T[\tilde{l}_2]\right) \tag{5}$$

where $\boldsymbol{u}_j$ are the columns of $\boldsymbol{U}$ and $[\boldsymbol{\Xi}] = \sum_{j=1}^{n_2}\frac{1}{\sigma_j^2 + \lambda}[\boldsymbol{H}_s^T]\boldsymbol{u}_j\boldsymbol{u}_j^T[\boldsymbol{H}_s]$. The result implies that, by separating the operations that involve encrypted data from those that do not, we can reduce the dimension of the matrix inversion from $(p+\ell) \times (p+\ell)$ to $\ell \times \ell$. Using the Newton-Schulz iterative algorithm [3] for matrix inversion, multiplications between $(p+\ell) \times (p+\ell)$ matrices reduce to multiplications between $\ell \times \ell$ matrices, which results in a quadratic reduction in the computation cost. Moreover, it is known that Newton-Schulz type methods can fail with large matrices [15], thus reducing the size of the matrix greatly improves the stability. It is notable that when $\ell = 1$ or $2$, the matrix inverse reduces to a scalar inverse, and can be more easily calculated with Goldschmidt's division algorithm [14].

Based on having the freedom to choose $\boldsymbol{u}_{p+1}, \ldots, \boldsymbol{u}_{n_2}$ because $\sigma_{p+1} = \cdots = \sigma_{n_2} = 0$, $[\boldsymbol{\Xi}]$ and $\hat{\boldsymbol{f}}$ can be further simplified as

---

**Algorithm 1** Training Ridge Regression with Encrypted Private Variable

---

**Input:** unencypted data $\boldsymbol{X}_{(-s)}$, encrypted private variable $[\boldsymbol{h}_s]$, encrypted target variable $[\boldsymbol{l}]$

**Output:** two ciphertexts $[\boldsymbol{\omega}_{(-s)}], [\omega_s]$ such that the weight vector of Ridge regression is $\boldsymbol{\omega}^T = (\boldsymbol{\omega}_{(-s)}^T \quad \omega_s)$

$\ll$*Pre-computations*$\gg$

[-] Calculate matrix $\boldsymbol{A}^{-1} = (\boldsymbol{X}_{(-s)}\boldsymbol{X}_{(-s)}^T + \lambda\boldsymbol{I}_{n_2})^{-1}$ using the collected user data $\mathbf{X}_{(-s)}$

[-] Obtain orthogonal matrices $\boldsymbol{U} = (\boldsymbol{u}_1 \cdots \boldsymbol{u}_p), \boldsymbol{V} = (\boldsymbol{v}_1 \dots \boldsymbol{v}_p)$ and diagonal entries $\boldsymbol{\sigma} = (\sigma_1, \cdots, \sigma_p)$ using the SVD of $\boldsymbol{X}_{(-s)}$

$\ll$*Computations on Encrypted data*$\gg$

[-] $[\boldsymbol{c}_1] \leftarrow \mathsf{MatVecProd}(\boldsymbol{V}, \frac{\boldsymbol{\sigma}}{\boldsymbol{\sigma}^2+\lambda}, \boldsymbol{U}, [\boldsymbol{l}]); [\boldsymbol{c}_2] \leftarrow \mathsf{MatVecProd}(\boldsymbol{V}, \frac{\boldsymbol{\sigma}}{\boldsymbol{\sigma}^2+\lambda}, \boldsymbol{U}, [\boldsymbol{h}_s]);$

[-] $[\boldsymbol{c}_3] \leftarrow \mathsf{MatVecProd}(\boldsymbol{U}, \frac{\boldsymbol{\sigma^2}}{\boldsymbol{\sigma}^2+\lambda}, \boldsymbol{U}, [\boldsymbol{l}]); [\boldsymbol{c}_4] \leftarrow \mathsf{MatVecProd}(\boldsymbol{U}, \frac{\boldsymbol{\sigma^2}}{\boldsymbol{\sigma}^2+\lambda}, \boldsymbol{U}, [\boldsymbol{h}_s]);$

[-] $[\boldsymbol{c}_5] \leftarrow ([\boldsymbol{h}_s] - [\boldsymbol{c}_4])/\lambda; [\boldsymbol{c}_6] \leftarrow \mathsf{InnerProduct}([\boldsymbol{c}_5], [\boldsymbol{l}]); [\boldsymbol{c}_7] \leftarrow \mathsf{Inv}(\mathsf{InnerProduct}([\boldsymbol{c}_5], [\boldsymbol{h}_s]) + 1)$

[-] Output $[\boldsymbol{\omega}_{(-s)}] \leftarrow [\boldsymbol{c}_1] - [\boldsymbol{c}_2] \cdot [\boldsymbol{c}_6] \cdot [\boldsymbol{c}_7]$ and $[\omega_s] \leftarrow [\boldsymbol{c}_6] \cdot [\boldsymbol{c}_7]$

---

$$[\boldsymbol{\Xi}] = \sum_{j=1}^{p+\ell} \frac{1}{\sigma_j^2 + \lambda}[\boldsymbol{H}_s^T]\boldsymbol{u}_j\boldsymbol{u}_j^T[\boldsymbol{H}_s],$$

$$\hat{\boldsymbol{f}} = [\tilde{\boldsymbol{l}}_2] - \lambda\boldsymbol{A}^{-1}[\tilde{\boldsymbol{l}}_2] + \lambda\left(\sum_{j=1}^{p+\ell} \frac{\boldsymbol{u}_j\boldsymbol{u}_j^T[\boldsymbol{H}_s]}{\sigma_j^2 + \lambda}\right)(\boldsymbol{I}_s + [\boldsymbol{\Xi}])^{-1}\left(\sum_{j=1}^{p+\ell} \frac{1}{\sigma_j^2 + \lambda}[\boldsymbol{H}_s^T]\boldsymbol{u}_j\boldsymbol{u}_j^T[\tilde{\boldsymbol{l}}_2]\right).$$

(6)

by choosing

$$\hat{\boldsymbol{u}}_{p+k} = \boldsymbol{h}_k - \sum_{i=1}^{p+k-1} \boldsymbol{u}_i(\boldsymbol{u}_i^T\boldsymbol{h}_k), \quad \boldsymbol{u}_{p+k} = \hat{\boldsymbol{u}}_{p+k}/\|\hat{\boldsymbol{u}}_{p+k}\|, \quad k = 1, ..., \ell.$$

**Algorithm** Algorithm 1 describes the procedure of training ridge regression, assuming that there is one private variable. For convenience, our method was described as obtaining the ridge estimate, but to infer the test data, the algorithm is slightly modified to output the weight vector. We assume that all $n_2$ samples of private variables are encoded into a single ciphertext to enable SIMD operations. If the number of samples is larger than the maximum number of samples which can be packed into a ciphertext, we can naturally split the samples into multiple ciphertexts. Excluding addition and multiplication operations for the ciphertext, the algorithm consists of three functions: InnerProduct, Inv, and MatVecProduct. InnerProduct returns a ciphertext in which the values of all plaintext slots are the same as the inner product of the two vectors. Inv evaluates Goldschmidt's division algorithm stated in Algorithm 1; however, in the case where $s \geq 3$, it evaluates an iterative method for matrix inversion. MatVecProduct evaluates efficient matrix-vector multiplication using SVD of the input matrix. The detailed algorithms of these functions are provided in the Appendix.

**Extension to nonlinear models** A limitation of our framework is that it can only model the linear relationship between the input variables and target variable. However, when partially encrypting private variables, our method has the capacity to model the nonlinear relationship between variables using kernel methods. The formulation opf the extension, and some experimental results are provided in the Appendix.

## 4 Theoretical Framework

This section provides the core theory behind our proposed methodology. We provide a generalization error bound for the proposed method.

Consider a classification task with an input space $\mathcal{X}$, an output space $\mathcal{Y} = \{0, 1\}$, and hypothesis space $\mathcal{Z} = \mathcal{X} \times [0, 1]$. Let $S = \{\mathbf{z}_i = (\mathbf{x}_i, y_i) \in \mathcal{Z} : i = 1, ..., N_s\}$ represent the given open labelled data of $N_s$ samples. In our classification settings, we have two distinct distributions: $(\mathbf{x}, y) \sim \mathcal{D}_S$

according to a sample distribution $\mathcal{D}_S$ over $\mathcal{Z}$ with $y \in \mathcal{Y}$ and $(\mathbf{x}, p) \sim \mathcal{D}_T$ according to a target distribution $\mathcal{D}_T$ over $\mathcal{Z}$ with $p \in [0, 1]$.

Utilizing these concepts and following the notations in [33], a hypothesis $h : \mathcal{Z} \to \Re$ is a scoring function such that we assign to each point $\mathbf{x}$ the class label of the maximum score $h(\mathbf{x}, y)$, that is, $\arg\max_{y \in \mathcal{Y}} h(\mathbf{x}, y)$. The *margin* $\xi_h(\mathbf{z})$ at a sample example $\mathbf{z} = (\mathbf{x}, y) \sim \mathcal{D}_S$ is then defined by

$$\xi_h(\mathbf{z}) = h(\mathbf{x}, y) - \max_{y' \neq y} h(\mathbf{x}, y')$$

Thus, $h$ misclassifies $\mathbf{z}$ iff $\xi_h(\mathbf{z}) \leq 0$.

The generalization error (or risk) of a hypothesis $h \in \mathcal{H} := \{h : \mathcal{Z} \to \Re\}$ in a sample distribution $\mathcal{D}_S$ and a true target distribution $\mathcal{D}_T$ are defined, respectively, by

$$\mathcal{R}_{\mathcal{D}_S}(h) = \mathbb{E}_{\mathbf{z} \sim \mathcal{D}_S}[1_{\xi_h(\mathbf{z}) \leq 0}], \qquad \mathcal{R}_{\mathcal{D}_T}(h) = \mathbb{E}_{\mathbf{z} \sim \mathcal{D}_T}[1_{\xi_h(\mathbf{z}) \leq 0}]$$

We next define the $\mathcal{H}$-divergence (short version of $\mathcal{H}\Delta\mathcal{H}$-divergence) as in [2] which measures the discrepancy between two different distributions as in by

$$D_{\mathcal{H}}(\mathcal{D}_S, \mathcal{D}_T) = \sup_{h, h' \in \mathcal{H}} |\mathcal{R}_{\mathcal{D}_S}(h') - \mathcal{R}_{\mathcal{D}_T}(h)|$$

We are now ready to derive the following generalization error risk in our classification setting.

**Lemma 1.** *Let $\mathcal{D}_S$ and $\mathcal{D}_T$ be the sample and the true target distributions, respectively. Then for any hypothesis $h \in \mathcal{H}$, the following inequality holds:*

$$\mathcal{R}_{\mathcal{D}_T}(h) \leq \min_{h' \in \mathcal{H}} \mathcal{R}_{\mathcal{D}_S}(h') + D_{\mathcal{H}}(\mathcal{D}_S, \mathcal{D}_T) \tag{7}$$

As a result, the upper bound of empirical expected target error can be decomposed into two parts: The first term is the empirical source error. In our framework, we try to minimize this term by training a teacher classification model with $\mathcal{D}_1$ in the first step. Then the second term is the $\mathcal{H}$-divergence between the $\mathcal{D}_S$ and $\mathcal{D}_T$ which implies that we should reduce the gap between two distributions to achieve better performance. The mean matching step in our framework corresponds to this part, which is a simplified version of kernel mean matching [17].

We next present the following margin bound for classification in the probably approximately correct (PAC) learning framework.

**Theorem 2.** *Let $\mathcal{D}_S$ and $\mathcal{D}_T$ be the sample and the true target distributions, respectively. Then, for any $\delta > 0$, with probability at least $1 - \delta$, the following classification generalization bound holds for all hypothesis $h \in \mathcal{H}_\rho = \{(\mathbf{x}, y) \to \boldsymbol{\beta} \cdot (y\mathbf{x}) : \|\boldsymbol{\beta}\|_2 \leq 1/\rho, \ \|\mathbf{x}\|_2 \leq r\}$:*

$$
\begin{aligned}
\mathcal{R}_{\mathcal{D}_T}(h) \quad \leq \quad & \frac{1}{N} \sum_{i=1}^{N} \log_{e_0} \left(1 + e^{-2y_i \boldsymbol{\beta} \cdot \mathbf{x}_i}\right) + D_{\mathcal{H}_\rho}(\mathcal{D}_S, \mathcal{D}_T) \\
& + \frac{16r}{\rho\sqrt{N}} + \sqrt{\frac{\log \log_2 \frac{4r}{\rho}}{N}} + \sqrt{\frac{\log \frac{2}{\delta}}{2N}}
\end{aligned} \tag{8}
$$

*where $e_0 = \log(1 + 1/e)$.*

## 5 Experiments

In this section, we evaluate our method using various real-world datasets. Through experiments, we argue that our method achieves better classification results compared to the existing methods with a shorter computation time. In addition, we verify that even when the distributions of published and encrypted data are very different, our method can properly correct the difference and maintain the classification performance.

**Datasets** We used five widely used classification datasets from the UCI data repository: The adult income dataset (**Adult**), bank marketing dataset (**Bank**), Wisconsin Breast Cancer dataset (**Cancer**), Pima Indians Diabetes dataset (**Diabetes**), and Australian Credit Approval (**Credit**) dataset [11]. **Adult** and **Bank** datasets are commonly used in fair machine learning studies, thus, we treat variables that can induce social bias, such as gender and age, like private variables. For the other datasets, the variable that had the greatest impact on the classification performance was selected as a private variable. In practical applications, data owners can flexibly set private variables according to their security standards. The explanation for each dataset is provided in the Appendix.

Table 1: Classification Results on five datasets.

| Dataset | Accuracy (%) | | | Computation time (sec) | | | Time per iteration (sec) | |
|---|---|---|---|---|---|---|---|---|
| | LRHE | Ours | Ours-grad | LRHE | Ours | Ours-grad | LRHE | Ours-grad |
| **Adult** | 81.913 | **83.123** | 82.759 | 306.061 | **102.25** | 1488.65 | 58.372 | **55.786** |
| **Bank** | 89.712 | 89.823 | **89.934** | 248.854 | **82.315** | 1095.44 | 47.528 | **43.184** |
| **Cancer** | 86.957 | **90.580** | 90.580 | 225.286 | **73.392** | 968.019 | 42.419 | **37.791** |
| **Diabetes** | 71.429 | 75.325 | **76.623** | 184.733 | **61.183** | 700.522 | 35.038 | **30.927** |
| **Credit** | 97.794 | 98.453 | **98.529** | 201.763 | **65.344** | 828.833 | 38.415 | **34.757** |

**Experimental Setup**   All the experiments were performed on a machine equipped with 40 threads of an Intel Xeon E-2660 v3 @2.60GHz CPU processor. We implemented step 1 of our framework with Python 3.6.3, using the LR module in the scikit-learn library. Other steps were implemented with C++, using HEAAN v1.1 [7] for HE. HEAAN is an implementation of the CKKS scheme; a detailed description of the scheme and its parameters are provided in the Appendix. We used privacy-preserving LR depicted in [23] (**LRHE**) as a baseline comparison method because it is the only MLED training model that works within a practical computation time. We do not compare our method to the HE-friendly SVM model reported in [36], because their model assumes that the data owners pre-compute the kernel matrix, which can leak information about the model. Comparing our model to DNN models reported in [29] is not of our interest because their model has a latency that is too high. Note that we trained **LRHE** with all variables encrypted because **LRHE** hardly benefits from partial encryption, as mentioned in Chapter 2. Indeed, encrypting only private variables for LRHE resulted in time savings of less than 1.2%.

For each dataset, we randomly sampled 20% as test samples, and 20% of the other 80% were treated as plaintext data, which were used for the training of step 1. We encrypted the private variables of the remaining 60% and used them for steps 2 and 3. For CKKS parameters, we used $N = 2^{16}, q_L = 2^{1200}$, and $P = 2^{40}$. The sigmoid function approximation degree for **LRHE** was set to 3 because increasing the degree results in a larger multiplicative depth and less possible number of gradient descents with LHE. In addition, we observed that increasing the degree up to 7 did not significantly affect the performance of the model. The learning rate for **LRHE** was chosen in $\{0.001, 0.0001, 0.00001\}$ to achieve the best classification performance.

**Results**   We compared the methods in terms of their classification accuracy and computation time. The results of the experiments are summarized in Table 1. For all datasets, **Ours** achieved higher accuracy and lower latency compared to **LRHE**. In particular, regarding the computation time, **Ours** can reduce latency by 66-68% through efficient computation using private variables. Although the performance of LR using plaintext is not inferior to ours, the performance of **LRHE** degrades owing to the sigmoid approximation and limited iterations of gradient descent. On the other hand, because **Ours** is free from parameter selection, its performance does not decrease compared to the same operation using plaintext.

As an ablation study, we additionally trained step 3 of **Ours** using gradient descent, with all variables encrypted (denoted as **Ours-grad** in Table 1). Because the multiplicative depth per iteration of **Ours-grad** is lower than that of **LRHE** owing to not using the sigmoid approximation, we trained **Ours-grad** for eight iterations. The learning rate was set in $\{0.001, 0.0001, 0.00001\}$.The procedure that requires the most computation time in **Ours-grad** is the matrix-matrix multiplication, which causes **Ours-grad** to have a greater latency compared to **LRHE**. However, the computation time per iteration of **LRHE** is higher than that of **Ours-grad**, and the result of the matrix-matrix multiplication can be reused through the iterations after calculating once. Therefore, it is implied that as the number of iterations increases, **Ours-grad** will be more efficient than **LRHE**. In addition, the classification performance of **Ours-grad** is similar to or better than that of **Ours**, which means that **Ours-grad** converges better by allowing more iterations than **LRHE**. Therefore, even in situations where the partial encryption of private variables is limited, our method has an advantage over the existing methods.

**Effectiveness of mean matching**   In this section, we verify that the mean matching in step 2 of our method is a simple but effective way to mitigate performance degradation due to the difference between distributions of unencrypted and encrypted data. Here, we do not randomly divide the dataset,

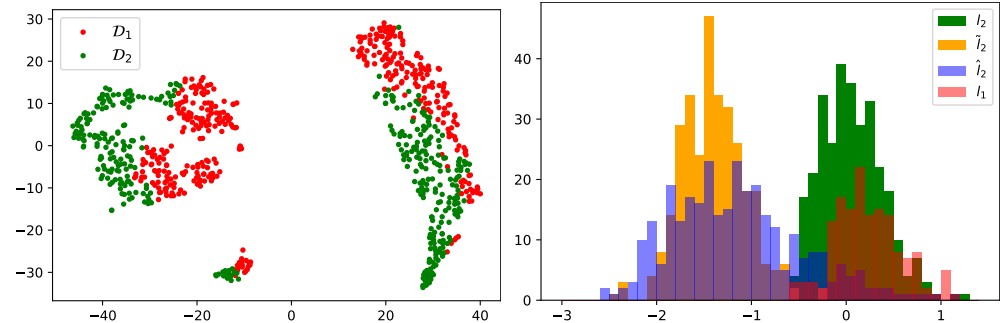

(a) Visualization of $\mathcal{D}_1$ and $\mathcal{D}_2$ with different distributions.

(b) Logit distribution of $\boldsymbol{l}_1$ (red), $\boldsymbol{l}_2$ (green), $\tilde{\boldsymbol{l}}_2$ (orange). The blue histogram refers to the distribution of logit of $X_2$ when trained with $X_2$.

Figure 3: Effectiveness of Mean Matching

Table 2: Ablation study for mean matching. **Criteria** refers to the criteria for dividing the datasets.

| **Dataset** | **Criteria** | | Accuracy (%) | | |
| | | **LR with $X_1$** | **Ours-without mean matching** | **Ours** | **LRHE** |
|---|---|---|---|---|---|
| **Adult** | $X_1$ : 'Marital' = 0 
 $X_2$ : 'Marital' = 1 | 63.330 | 63.330 | 69.471 | 68.690 |
| **Diabetes** | $X_1$ : 'Glucose' > 117 
 $X_2$ : 'Glucose' $\leq$ 117 | 29.487 | 29.487 | 80.769 | 80.769 |
| **Credit** | $X_1$ : 'A4' = 0 
 $X_2$ : 'A4' = 1 | 20.833 | 20.833 | 79.167 | 79.167 |

instead divide it referring to the value of a certain variable. **Diabetes** dataset, for example, was divided according to whether the value of 'Glucose' variable was greater than (or less than) its median. Figure 3(a) plots the dimension reduction result for **Diabetes** dataset using t-stochastic neighborhood embedding and confirms that the two subsets ($X_1$ and $X_2$) have very separate distributions compared to each other. We sampled 20% of the entire dataset from $X_1$ and trained step 1 with the samples. 80% of $X_2$ was used for steps 2 and 3, with the remaining 20% of $X_2$ used as the test samples.

Table 2 summarizes the experimental results for three datasets: **Adult**, **Diabetes** and **Credit**. For the other two, there were no variables that could properly divide them. The test accuracy of the LR model trained with $X_1$ was lower than that of the **LRHE** trained with $X_2$ because the domains of $X_1$ and $X_2$ were different, and the test set was sampled from $X_2$. The performance of **Ours** without mean matching was not different from that of the LR with $X_1$. However, with mean matching the classification accuracy of **Ours** was raised to the same level as that of **LRHE**. Even when the accuracy of LR with $X_1$ was 50% or lower than that of **LRHE**, applying mean matching completely recovered the classification performance of **Ours**.

To verify why the mean matching of our method works well, in Figure 3(b), we plotted the histogram of logits inferred by LR models. In the figure, the red and blue histograms represent the distributions of logit of $X_1$ for LR learned with $X_1$ ($=\boldsymbol{l}_1$) and logit of $X_2$ for LR learned with $X_2$, respectively. Thus, the blue histogram can be seen as the distribution of the logit of $X_2$ when it is classified "correctly". However, because the model is biased toward the distribution of $X_1$ when trained with $X_1$, the distribution of the logit of $X_2$ ($=\boldsymbol{l}_2$) is also biased toward the distribution of the logit of $X_1$, as shown in the green histogram. Therefore, mean matching plays the role of shifting the biased distribution (orange histogram) to fit the correct distribution.

## 6    Conclusion

We proposed an efficient HE-friendly classification method, that protects both the users' private information and secrecy of the model. We trained a ridge regression model for the logit instead of logistic regression, which has a closed-form solution and is free from parameter search. To extract

logit values from a binary label, we trained a teacher model on unencrypted data that can output logit, similar to that of knowledge distillation. Owing to encrypting only private variables that require a high level of security instead of encrypting all information, our method can achieve higher efficiency and training stability. Our algorithm is HE scheme-free; it can bring efficiency when implemented with any widely used HE schemes.

Although our study aims to extend the existing machine learning method in the privacy-preserving direction, our model can be corrupted by malicious participants because our security model is limited to semi-honest, non-colluding participants. Designing a secure system for weaker security assumptions should be conducted in future studies.

## Acknowledgments and Disclosure of Funding

This research was supported by Basic Science Research Program through the National Research Foundation of Korea (NRF) funded by the Ministry of Science and Technology Information and Communication (2019R1A2C2002358)

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
