# Appendix for Parameter-free HE-friendly Logistic Regression

## 1 Additional Details on Theoretical Framework

**Lemma 1** *Let $\mathcal{D}_S$ and $\mathcal{D}_T$ be the sample and the true target distributions, respectively. Then for any hypothesis $h \in \mathcal{H}$, the following inequality holds:*

$$\mathcal{R}_{\mathcal{D}_T}(h) \leq \min_{h' \in \mathcal{H}} \mathcal{R}_{\mathcal{D}_S}(h') + D_{\mathcal{H},\mathcal{H}}(\mathcal{D}_S, \mathcal{D}_T) \tag{1}$$

**Proof 1** *For a given $h \in \mathcal{H}$,*

$$\begin{aligned}
\mathcal{R}_{\mathcal{D}_T}(h) &= \mathcal{R}_{\mathcal{D}_S}(h') + \mathcal{R}_{\mathcal{D}_T}(h) - \mathcal{R}_{\mathcal{D}_S}(h') \\
&\leq \mathcal{R}_{\mathcal{D}_S}(h') + D_{\mathcal{H}}(\mathcal{D}_S, \mathcal{D}_T)
\end{aligned}$$

*which holds for all $h' \in \mathcal{H}$. Therefore, the result follows.* ¶

Following the notations and definitions in Mohri et al. [2018], let $S = \{\mathbf{z}_i = (\mathbf{x}_i, y_i) \sim \mathcal{D}_S : i = 1, ..., N\}$ represent the class labeled data of $N$ samples. Then the empirical Rademacher complexity of $\mathcal{H}$ with respect to $S$ is the random variable

$$\hat{\mathfrak{R}}_{\mathcal{D}_S}(\mathcal{H}) = \mathbb{E}_{\boldsymbol{\sigma}} \left[ \sup_{h \in \mathcal{H}} \frac{1}{N} \sum_{i=1}^{N} \sigma_i h(\mathbf{z}_i) \right] \tag{2}$$

where $\boldsymbol{\sigma} = \{\sigma_1, ..., \sigma_N\}$ are independent uniform $\{\pm 1\}$-valued Rademacher random variables. The *Rademacher complexity* of $\mathcal{H}$ is the expectation of the empirical Rademacher complexity over all samples of size $N$:

$$\mathfrak{R}_N(\mathcal{H}) = \mathbb{E}_{\mathcal{D}_S}[\hat{\mathfrak{R}}_{\mathcal{D}_S}(\mathcal{H})] = \mathbb{E}_{S\boldsymbol{\sigma}} \left[ \sup_{h \in \mathcal{H}} \frac{1}{N} \sum_{i=1}^{N} \sigma_i h(\mathbf{z}_i) \right] \tag{3}$$

**Theorem 1** *Let $\mathcal{D}_S$ and $\mathcal{D}_T$ be the sample and the true target distributions, respectively. Then, for any $\delta > 0$, with probability at least $1 - \delta$, the following classification generalization bound holds for all hypothesis $h \in \mathcal{H}_\rho = \{(\mathbf{x}, y) \to \boldsymbol{\omega} \cdot (y\mathbf{x}) : \|\boldsymbol{\omega}\|_2 \leq 1/\rho, \|\mathbf{x}\|_2 \leq r\}$:*

$$\mathcal{R}_{\mathcal{D}_T}(h) \leq \frac{1}{N} \sum_{i=1}^{N} \log_{e_0}\left(1 + e^{-2y_i \boldsymbol{\omega} \cdot \mathbf{x}_i}\right) + D_{\mathcal{H}_\rho}(\mathcal{D}_S, \mathcal{D}_T) \tag{4}$$

$$+ \frac{16r}{\rho\sqrt{N}} + \sqrt{\frac{\log \log_2 \frac{4r}{\rho}}{N}} + \sqrt{\frac{\log \frac{2}{\delta}}{2N}} \tag{5}$$

*where $e_0 = \log(1 + 1/e)$.*

**Proof 2** *We define $\Lambda(\mathcal{H}_1)$ for a scoring function $h \in \mathcal{H}_1$ by*

$$\Lambda(\mathcal{H}_1) = \{\mathbf{x} \mapsto h(\mathbf{x}, y) : y \in \mathcal{Y}, \ h \in \mathcal{H}_1\} = \{\mathbf{x} \mapsto y(\boldsymbol{\omega} \cdot \mathbf{x}) : y \in \mathcal{Y}, \ \|\boldsymbol{\omega}\|_2 \leq 1\}$$

*Then for any $0 < \rho < 2r$, by Theorem 9.2 and Theorem 13.2, 13.4 in Mohri et al. [2018] and some slight modifications adapted to our classification setting, we have the following general margin bound of $\mathcal{R}_{\mathcal{D}_S}(h)$:*

$$\mathcal{R}_{\mathcal{D}_S}(h) \leq \frac{1}{N}\sum_{i=1}^{N} 1_{\xi_h(\mathbf{z}_i) \leq \rho} + \frac{8}{\rho}\mathfrak{R}_N(\Lambda(\mathcal{H}_1)) + \sqrt{\frac{\log\log_2 \frac{4r}{\rho}}{N}} + \sqrt{\frac{\log\frac{2}{\delta}}{2N}}$$

*where $e_0 = \log(1 + 1/e)$.*

*We next derive a margin upper bound of $\mathfrak{R}_N(\Lambda(\mathcal{H}_1))$. Since $|\boldsymbol{\omega} \cdot y\mathbf{x}| \leq \|\boldsymbol{\omega}\|_2 \|y\mathbf{x}\|_2 \leq r$ by the Cauchy–Schwarz inequality, we have*

$$\hat{\mathfrak{R}}_N(\Lambda(\mathcal{H}_1)) = \frac{1}{N}\mathbb{E}_\sigma\left[\sup_{\|\boldsymbol{\omega}\|_2 \leq 1, y \in \mathcal{Y}} \boldsymbol{\omega} \cdot \sum_{i=1}^N y\sigma_i\mathbf{x}_i\right] = \frac{1}{N}\mathbb{E}_\sigma\left[\sup_{y \in \mathcal{Y}}\|\sum_{i=1}^N y\sigma_i\mathbf{x}_i)\|_2\right]$$

$$\leq \frac{1}{N}\sum_{y \in \mathcal{Y}}\mathbb{E}_\sigma\left[\|\sum_{i=1}^N y\sigma_i\mathbf{x}_i\|_2\right] \leq \frac{1}{N}\sum_{y \in \mathcal{Y}}\sqrt{\mathbb{E}_\sigma\left[\|\sum_{i=1}^N y\sigma_i\mathbf{x}_i\|_2^2\right]}$$

$$\leq \frac{1}{N}\sum_{y \in \mathcal{Y}}\sqrt{\sum_{i=1}^N \|y\mathbf{x}_i\|_2^2} \leq \frac{2r}{\sqrt{N}}$$

*where the third inequality holds by definition of the dual norm. Since*

$$\sum_{y \in \mathcal{Y}}\exp(\frac{h(\mathbf{x}_i, y) - h(\mathbf{x}_i, y_i)}{\rho}) = \sum_{y \in \mathcal{Y}}\exp(\frac{y\boldsymbol{\omega} \cdot \mathbf{x}_i - y_i\boldsymbol{\omega} \cdot \mathbf{x}_i}{\rho})$$

$$= \exp(\frac{\boldsymbol{\omega} \cdot \mathbf{x}_i - y_i\boldsymbol{\omega} \cdot \mathbf{x}_i}{\rho}) + \exp(\frac{-\boldsymbol{\omega} \cdot \mathbf{x}_i - y_i\boldsymbol{\omega} \cdot \mathbf{x}_i}{\rho}) = 1 + \exp(\frac{-2y_i\boldsymbol{\omega} \cdot \mathbf{x}_i}{\rho})$$

*if we let $\tilde{\boldsymbol{\omega}} = \boldsymbol{\omega}/\rho$, then $\|\tilde{\boldsymbol{\omega}}\| \leq 1/\rho$*

$$\frac{1}{N}\sum_{i=1}^N 1_{\xi_h(\mathbf{z}_i) \leq \rho} \leq \frac{1}{N}\sum_{i=1}^N \log_{e_0}\left(\sum_{y \in \mathcal{Y}} e^{\frac{h(\mathbf{x}_i, y_i) - h(\mathbf{x}_i, y)}{\rho}}\right) = \frac{1}{N}\sum_{i=1}^N \log_{e_0}\left(1 + \exp(-2y_i\tilde{\boldsymbol{\omega}} \cdot \mathbf{x}_i)\right)$$

*Therefore we have*

$$\mathcal{R}_{\mathcal{D}_S}(h) \leq \frac{1}{N}\sum_{i=1}^N \log_{e_0}\left(1 + \exp(-2y_i\tilde{\boldsymbol{\omega}} \cdot \mathbf{x}_i)\right) + \frac{16r}{\rho\sqrt{N}} + \sqrt{\frac{\log\log_2 \frac{4r}{\rho}}{N}} + \sqrt{\frac{\log\frac{2}{\delta}}{2N}}$$

*and the result follows from Lemma 1. ¶*

Notice that the first term of the right-hand side of Eq. (4) is the loss function for the logistic regression where the conditional probability takes the form of logit function:

$$\Pr[y = 1|x] = \frac{1}{1 + e^{-2\boldsymbol{\omega} \cdot \mathbf{x}}}$$

As a result, the upper bound of the empirical target error is composed with the loss function for the logistic regression over the sample distribution $\mathcal{D}_s$ and the $\mathcal{H},\mathcal{H}$-divergence between two distributions. In our framework by step 2 and step 3, we tried to mitigate the difference between the two distributions. Through Figure 3-(b) and empirical results in Table 2, we have shown that our framework works well, even though the underlying distribution of $\mathcal{D}_S$ and $\mathcal{D}_T$ are different.

## 2 Details on Ridge Regression with Private Variables

### 2.1 Regression with one Encrypted Private Variable

We start with the following multivariate linear regression with one dependent variable $Y$, $p$- independent *non-private* variables $X_1, X_2, \ldots, X_p$, and one *private variable* $X_s$:

$$Y = \omega_0 + \omega_1 X_1 + \omega_2 X_2 + \cdots + \omega_p X_p + \omega_s X_s + \varepsilon \tag{6}$$

where $\boldsymbol{\omega}_I = (\omega_0, \omega_1, ..., \omega_p, \omega_s)^T$ consists of regression coefficients to be estimated and $\varepsilon$ is error term. For $i = 1, ...n$, $(x_{i1}, ..., x_{ip}, x_{is}, y_i)$ denotes the $i$-th observations of $X_1, X_2, ..., X_p, X_s$, and $Y$. We assume that $n > p + 1$, which means that there are more observations than the number of independent variables.

### 2.1.1 Homomorphic encryption of a private variable

Let $h_i = h(x_{is})$ be the fully homomorphic encryption of a private variable $x_{is}$, then the data transferred to the server is $(x_{i1}, ..., x_{ip}, h(x_{is}), y_i)$ for $i = 1, ...n$. For the convenience, from now on, operations between a plaintext and a ciphertext, or between ciphertexts, are denoted as those between plaintexts.

Using centered inputs by replacing each $x_{ij}$ with $x_{ij} - \bar{x}_j$, each $h(x_{is})$ with $h(x_{is}) - \bar{h}_s$, and estimating $\omega_0$ by $\bar{y} = \sum_{i=1}^{n} y_i$, (6) becomes a regression model without intercept as

$$y_i = \omega_1 x_{i1} + \omega_2 x_{i2} + \cdots + \omega_p x_{ip} + \omega_s h(x_{is}) + \varepsilon_i, \quad i = 1, ..., n$$
$$\boldsymbol{y} = \boldsymbol{X}\boldsymbol{\omega} + \boldsymbol{\varepsilon}$$

$$= \underbrace{\begin{bmatrix} x_{11} & \cdots & x_{1p} & h(x_{1s}) \\ \vdots & \ddots & \vdots & \vdots \\ x_{i1} & \cdots & x_{ip} & h(x_{is}) \\ \vdots & \ddots & \vdots & \vdots \\ x_{n1} & \cdots & x_{np} & h(x_{ns}) \end{bmatrix}}_{X} \underbrace{\begin{bmatrix} \omega_1 \\ \vdots \\ \omega_p \\ \omega_s \end{bmatrix}}_{\omega} + \underbrace{\begin{bmatrix} \varepsilon_1 \\ \varepsilon_2 \\ \vdots \\ \varepsilon_n \end{bmatrix}}_{\varepsilon} \tag{7}$$

### 2.1.2 Ridge estimate without matrix inverse on an encrypted variable

By adding a regularization term to an error function in order to control over-fitting, the total error function to be minimized takes the form

$$\mathcal{E}(\boldsymbol{\omega}) = \mathcal{E}_D(\boldsymbol{\omega}) + \mathcal{E}_W(\boldsymbol{\omega})$$
$$= \frac{1}{2}\sum_{i=1}^{N}(y_i - (\omega_1 x_{i1} + \omega_2 x_{i2} + \cdots + \omega_p x_{ip} + \omega_s h(x_{is})))^2 + \frac{\lambda}{2}\boldsymbol{\omega}^T\boldsymbol{\omega}. \tag{8}$$

The ridge regression solutions are then shown to be

$$\hat{\boldsymbol{\omega}}_{RLS} = (\boldsymbol{X}^T\boldsymbol{X} + \lambda\boldsymbol{I}_{p+1})^{-1}\boldsymbol{X}^T\boldsymbol{y} \tag{9}$$

where $\boldsymbol{I}_{p+1}$ is a $(p+1) \times (p+1)$-identity matrix. Note that the intercept $\omega_0$ is not regularized because of centering of the variables. The ridge estimate is therefore

$$\hat{\boldsymbol{f}} = \boldsymbol{X}(\boldsymbol{X}^T\boldsymbol{X} + \lambda\boldsymbol{I}_{p+1})^{-1}\boldsymbol{X}^T\boldsymbol{y} = \boldsymbol{X}\boldsymbol{X}^T(\boldsymbol{X}\boldsymbol{X}^T + \lambda\boldsymbol{I}_n)^{-1}\boldsymbol{y} \tag{10}$$

where $\boldsymbol{X}\boldsymbol{X}^T$ is an $n \times n$ matrix.
The last equality holds because $\boldsymbol{X}^T(\boldsymbol{X}\boldsymbol{X}^T + \lambda\boldsymbol{I}_n) = (\boldsymbol{X}^T\boldsymbol{X} + \lambda\boldsymbol{I}_{p+1})\boldsymbol{X}^T$, and therefore $(\boldsymbol{X}^T\boldsymbol{X} + \lambda\boldsymbol{I}_{p+1})^{-1}\boldsymbol{X}^T = \boldsymbol{X}^T(\boldsymbol{X}\boldsymbol{X}^T + \lambda\boldsymbol{I}_n)^{-1}$.

Since

$$\boldsymbol{X}\boldsymbol{X}^T = \sum_{i=1}^{p}\boldsymbol{x}_i\boldsymbol{x}_i^T + \boldsymbol{h}_s\boldsymbol{h}_s^T = \boldsymbol{X}_{(-s)}\boldsymbol{X}_{(-s)}^T + \boldsymbol{h}_s\boldsymbol{h}_s^T \tag{11}$$

where $\boldsymbol{h}_s = (h(x_{1s}), ..., h(x_{ns}))^T$ and $\boldsymbol{X}_{(-s)}$ is the other part of $\boldsymbol{X}$, by Sherman-Woodbury inversion formula, we have

$$(\boldsymbol{X}\boldsymbol{X}^T + \lambda\boldsymbol{I}_n)^{-1} = (\boldsymbol{X}_{(-s)}\boldsymbol{X}_{(-s)}^T + \lambda\boldsymbol{I}_n + \boldsymbol{h}_s\boldsymbol{h}_s^T)^{-1}$$
$$= (\boldsymbol{X}_{(-s)}\boldsymbol{X}_{(-s)}^T + \lambda\boldsymbol{I}_n)^{-1} - \frac{(\boldsymbol{X}_{(-s)}\boldsymbol{X}_{(-s)}^T + \lambda\boldsymbol{I}_n)^{-1}\boldsymbol{h}_s\boldsymbol{h}_s^T(\boldsymbol{X}_{(-s)}\boldsymbol{X}_{(-s)}^T + \lambda\boldsymbol{I}_n)^{-1}}{1 + \boldsymbol{h}_s^T(\boldsymbol{X}_{(-s)}\boldsymbol{X}_{(-s)}^T + \lambda\boldsymbol{I}_n)^{-1}\boldsymbol{h}_s}$$
$$= \boldsymbol{A}^{-1} - \frac{\boldsymbol{A}^{-1}\boldsymbol{h}_s\boldsymbol{h}_s^T\boldsymbol{A}^{-1}}{1 + \boldsymbol{h}_s^T\boldsymbol{A}^{-1}\boldsymbol{h}_s} \tag{12}$$

where $\boldsymbol{A} = \boldsymbol{X}_{(-s)}\boldsymbol{X}_{(-s)}^T + \lambda \boldsymbol{I}_n$. Using the singular vector decomposition (SVD) of $\boldsymbol{X}_{(-s)} = U\Sigma V^T$ where $U \in \Re^{n \times n}$ and $V \in \Re^{p \times p}$ are orothogonal matrices and $\Sigma \in \Re^{n \times p}$ are diagonal matrix with diagonal entries $\sigma_1 \geq .... \geq \sigma_p$, we have

$$\boldsymbol{A}^{-1} = (\boldsymbol{X}_{(-s)}\boldsymbol{X}_{(-s)}^T + \lambda \boldsymbol{I}_n)^{-1} = (U\Sigma\Sigma^T U^T + \lambda \boldsymbol{I}_n)^{-1} = U(\Sigma\Sigma^T + \lambda \boldsymbol{I}_n)^{-1}U^T \tag{13}$$

and letting $\sigma_{p+1} = \cdots = \sigma_n = 0$, we define the following terms

$$\xi = \boldsymbol{h}_s^T \boldsymbol{A}^{-1}\boldsymbol{h}_s = \boldsymbol{h}_s^T U(\Sigma\Sigma^T + \lambda \boldsymbol{I}_n)^{-1}U^T \boldsymbol{h}_s = \sum_{j=1}^n \boldsymbol{h}_s^T \boldsymbol{u}_j \frac{1}{\sigma_j^2 + \lambda} \boldsymbol{u}_j^T \boldsymbol{h}_s = \sum_{j=1}^n \frac{(\boldsymbol{h}_s^T \boldsymbol{u}_j)^2}{\sigma_j^2 + \lambda}$$

$$\eta = \boldsymbol{h}_s^T \boldsymbol{A}^{-1}\boldsymbol{y} = \boldsymbol{h}_s^T U(\Sigma\Sigma^T + \lambda \boldsymbol{I}_n)^{-1}U^T \boldsymbol{y} = \sum_{j=1}^n \frac{(\boldsymbol{h}_s^T \boldsymbol{u}_j)(\boldsymbol{u}_j^T \boldsymbol{y})}{\sigma_j^2 + \lambda}. \tag{14}$$

Then the ridge estimate becomes

$$\begin{aligned}
\hat{\boldsymbol{f}} &= \boldsymbol{X}\boldsymbol{X}^T(\boldsymbol{X}\boldsymbol{X}^T + \lambda \boldsymbol{I}_n)^{-1}\boldsymbol{y} = (\boldsymbol{X}\boldsymbol{X}^T + \lambda \boldsymbol{I}_n - \lambda \boldsymbol{I}_n)(\boldsymbol{X}\boldsymbol{X}^T + \lambda \boldsymbol{I}_n)^{-1}\boldsymbol{y} \\
&= \boldsymbol{y} - \lambda(\boldsymbol{X}\boldsymbol{X}^T + \lambda \boldsymbol{I}_n)^{-1}\boldsymbol{y} = \boldsymbol{y} - \lambda \boldsymbol{A}^{-1}\boldsymbol{y} + \lambda \frac{\boldsymbol{A}^{-1}\boldsymbol{h}_s\boldsymbol{h}_s^T \boldsymbol{A}^{-1}}{1 + \boldsymbol{h}_s^T \boldsymbol{A}^{-1}\boldsymbol{h}_s}\boldsymbol{y} \\
&= \boldsymbol{y} - \lambda \boldsymbol{A}^{-1}\boldsymbol{y} + \frac{\lambda \eta}{1 + \xi}\boldsymbol{A}^{-1}\boldsymbol{h}_s \\
&= \boldsymbol{y} - \lambda U(\Sigma\Sigma^T + \lambda \boldsymbol{I}_n)^{-1}U^T \boldsymbol{y} + \frac{\lambda \eta}{1 + \xi}U(\Sigma\Sigma^T + \lambda \boldsymbol{I}_n)^{-1}U^T \boldsymbol{h}_s \\
&= \boldsymbol{y} - \sum_{j=1}^n \frac{\lambda \boldsymbol{u}_j(\boldsymbol{u}_j^T \boldsymbol{y})}{\sigma_j^2 + \lambda} + \frac{\lambda \eta}{1 + \xi}\sum_{j=1}^n \frac{\boldsymbol{u}_j(\boldsymbol{u}_j^T \boldsymbol{h}_s)}{\sigma_j^2 + \lambda} \\
&= \sum_{j=1}^n \frac{\sigma_j^2}{\sigma_j^2 + \lambda}\boldsymbol{u}_j(\boldsymbol{u}_j^T \boldsymbol{y}) + \frac{\lambda \eta}{1 + \xi}\sum_{j=1}^n \frac{\boldsymbol{u}_j(\boldsymbol{u}_j^T \boldsymbol{h}_s)}{\sigma_j^2 + \lambda}
\end{aligned} \tag{15}$$

where $\boldsymbol{u}_j$ are the columns of $U \in \Re^{n \times n}$ and $\sum_{j=1}^n \boldsymbol{u}_j\boldsymbol{u}_j^T = \boldsymbol{I}_n$.

### 2.1.3 Fast Ridge estimate

The above computation involves $n$ summations. To simplify the computations, we notice that $U = [U_1, U_2] \in \Re^{n \times n}$ where $U_1 = [\boldsymbol{u}_1, ..., \boldsymbol{u}_p] \in \Re^{n \times p}$ and $U_2 = [\boldsymbol{u}_{p+1}, ..., \boldsymbol{u}_n] \in \Re^{n \times (n-p)}$ and $U_1 \perp U_2$. Using the reduced SVD and the fact that $\sigma_{p+1} = \cdots = \sigma_n = 0$, we have the freedom to choose $U_2$ as long as $U_1 \perp U_2$. Therefore, we choose $\boldsymbol{u}_{p+1}$ in such a way that $\boldsymbol{u}_j$ are orthogonal to $\boldsymbol{h}_s$ for all $j = p + 2, ..., n$ by letting $\boldsymbol{u}_{p+1}$ be the complement of the orthogonal projection of $\boldsymbol{h}_s$ onto $U_1$ as follows:

$$\hat{\boldsymbol{u}}_{p+1} = (\boldsymbol{I}_n - P_{U_1})\boldsymbol{h}_s = (\boldsymbol{I}_n - \sum_{i=1}^p \boldsymbol{u}_i\boldsymbol{u}_i^T)\boldsymbol{h}_s = \boldsymbol{h}_s - \sum_{i=1}^p \boldsymbol{u}_i(\boldsymbol{u}_i^T \boldsymbol{h}_s)$$

$$\boldsymbol{u}_{p+1} = \hat{\boldsymbol{u}}_{p+1}/\|\hat{\boldsymbol{u}}_{p+1}\|, \quad \boldsymbol{u}_{p+1}(\boldsymbol{u}_{p+1}^T \boldsymbol{h}_s) = \boldsymbol{h}_s - \sum_{i=1}^p \boldsymbol{u}_i(\boldsymbol{u}_i^T \boldsymbol{h}_s). \tag{16}$$

Then $\xi$ and $\eta$ can be simplified as

$$\eta = \sum_{j=1}^{p+1} \frac{\boldsymbol{h}_s^T \boldsymbol{u}_j}{\sigma_j^2 + \lambda} \boldsymbol{u}_j^T \boldsymbol{y} = \sum_{j=1}^{p} \frac{\boldsymbol{h}_s^T \boldsymbol{u}_j}{\sigma_j^2 + \lambda} \boldsymbol{u}_j^T \boldsymbol{y} + \frac{1}{\lambda} \boldsymbol{y}^T \boldsymbol{u}_{p+1} (\boldsymbol{u}_{p+1}^T \boldsymbol{h}_s)$$

$$= \sum_{j=1}^{p} \frac{(\boldsymbol{h}_s^T \boldsymbol{u}_j)(\boldsymbol{u}_j^T \boldsymbol{y})}{\sigma_j^2 + \lambda} + \frac{1}{\lambda} \boldsymbol{y}^T (\boldsymbol{h}_s - \sum_{j=1}^{p} \boldsymbol{u}_j (\boldsymbol{u}_j^T \boldsymbol{h}_s))$$

$$= \frac{1}{\lambda} \left[ -\sum_{j=1}^{p} \frac{\sigma_j^2 (\boldsymbol{h}_s^T \boldsymbol{u}_j)(\boldsymbol{u}_j^T \boldsymbol{y})}{\sigma_j^2 + \lambda} + \boldsymbol{y}^T \boldsymbol{h}_s \right]$$

$$\xi = \sum_{j=1}^{p+1} \frac{(\boldsymbol{h}_s^T \boldsymbol{u}_j)^2}{\sigma_j^2 + \lambda} = \sum_{j=1}^{p} \frac{(\boldsymbol{h}_s^T \boldsymbol{u}_j)^2}{\sigma_j^2 + \lambda} + \frac{1}{\lambda} \boldsymbol{h}_s^T \boldsymbol{u}_{p+1} (\boldsymbol{u}_{p+1}^T \boldsymbol{h}_s)$$

$$= \sum_{j=1}^{p} \frac{(\boldsymbol{h}_s^T \boldsymbol{u}_j)^2}{\sigma_j^2 + \lambda} + \frac{1}{\lambda} \boldsymbol{h}_s^T (\boldsymbol{h}_s - \sum_{j=1}^{p} \boldsymbol{u}_j (\boldsymbol{u}_j^T \boldsymbol{h}_s))$$

$$= \frac{1}{\lambda} \left[ -\sum_{j=1}^{p} \frac{\sigma_j^2 (\boldsymbol{h}_s^T \boldsymbol{u}_j)^2}{\sigma_j^2 + \lambda} + \boldsymbol{h}_s^T \boldsymbol{h}_s \right]. \tag{17}$$

Therefore the ridge estimate can be further simplified as

$$\hat{\boldsymbol{f}} = \sum_{i=1}^{p} \frac{\sigma_i^2}{\sigma_i^2 + \lambda} \boldsymbol{u}_i (\boldsymbol{u}_i^T \boldsymbol{y}) + \frac{\lambda \eta}{1 + \xi} \sum_{j=1}^{p+1} \frac{\boldsymbol{u}_j (\boldsymbol{u}_j^T \boldsymbol{h}_s)}{\sigma_j^2 + \lambda}$$

$$= \sum_{i=1}^{p} \frac{\sigma_i^2}{\sigma_i^2 + \lambda} \boldsymbol{u}_i (\boldsymbol{u}_i^T \boldsymbol{y}) + \frac{\lambda \eta}{1 + \xi} \left[ \sum_{j=1}^{p} \frac{\boldsymbol{u}_j (\boldsymbol{u}_j^T \boldsymbol{h}_s)}{\sigma_j^2 + \lambda} + \frac{1}{\lambda} (\boldsymbol{h}_s - \sum_{i=1}^{p} \boldsymbol{u}_i (\boldsymbol{u}_i^T \boldsymbol{h}_s)) \right]$$

$$= \sum_{i=1}^{p} \boldsymbol{u}_i \frac{\sigma_i^2}{\sigma_i^2 + \lambda} \boldsymbol{u}_i^T \boldsymbol{y} - \frac{\eta}{1 + \xi} \sum_{i=1}^{p} \boldsymbol{u}_i \frac{\sigma_i^2 (\boldsymbol{u}_i^T \boldsymbol{h}_s)}{\sigma_i^2 + \lambda} + \frac{\eta}{1 + \xi} \boldsymbol{h}_s \tag{18}$$

which involves only $p$ summations.

## 2.2 Regression with multiple Private Variables

Let $(x_{i1}, \ldots, x_{ip}, x_{is1}, \ldots, x_{is\ell}, y_i)$ be the i-th observations of $p$- independent *non-private* variables $X_1, X_2, \ldots, X_p$, and *private variable*s $X_{s1}, \ldots, X_{s\ell}$, and one dependent variable $Y$.

### 2.2.1 Homomorphic encryption of private variables

Let $h_{ij} = h_j(x_{isj})$ be the fully homomorphic encription of the private variable $x_{is1}, \ldots, x_{is\ell}$ where $(x_{i1}, \ldots, x_{ip}, h_1(x_{is1}), \cdots, h_\ell(x_{is\ell}), y_i)$ for $i = 1, \ldots n$.

Using centered inputs as above, regression model without intercept becomes as follows.

$$\boldsymbol{y} = \boldsymbol{X}\boldsymbol{\omega} + \boldsymbol{\varepsilon}$$

$$= \underbrace{\begin{bmatrix} x_{11} & \cdots & x_{1p} & h_1(x_{1s1}) & \cdots & h_\ell(x_{1s\ell}) \\ \vdots & \ddots & \vdots & \vdots & \ddots & \vdots \\ x_{i1} & \cdots & x_{ip} & h_1(x_{is1}) & \cdots & h_\ell(x_{is\ell}) \\ \vdots & \ddots & \vdots & \vdots & \ddots & \vdots \\ x_{n1} & \cdots & x_{np} & h_1(x_{ns1}) & \cdots & h_\ell(x_{ns\ell}) \end{bmatrix}}_{X} \underbrace{\begin{bmatrix} \omega_1 \\ \vdots \\ \omega_p \\ \omega_{s1} \\ \vdots \\ \omega_{s\ell} \end{bmatrix}}_{\omega} + \underbrace{\begin{bmatrix} \varepsilon_1 \\ \varepsilon_2 \\ \vdots \\ \varepsilon_n \end{bmatrix}}_{\varepsilon} \tag{19}$$

### 2.2.2 Ridge estimate on encrypted variables

The ridge regression solution and the ridge estimate are

$$
\begin{aligned}
\hat{\boldsymbol{\omega}}_{RLS} &= (\boldsymbol{X}^T\boldsymbol{X} + \lambda \boldsymbol{I}_{p+\ell})^{-1}\boldsymbol{X}^T\boldsymbol{y} = \boldsymbol{X}^T(\boldsymbol{X}\boldsymbol{X}^T + \lambda \boldsymbol{I}_n)^{-1}\boldsymbol{y} \\
\hat{\boldsymbol{f}} &= \boldsymbol{X}(\boldsymbol{X}^T\boldsymbol{X} + \lambda \boldsymbol{I}_{p+\ell})^{-1}\boldsymbol{X}^T\boldsymbol{y} = \boldsymbol{X}\boldsymbol{X}^T(\boldsymbol{X}\boldsymbol{X}^T + \lambda \boldsymbol{I}_n)^{-1}\boldsymbol{y} \\
&= \boldsymbol{y} - \lambda(\boldsymbol{X}\boldsymbol{X}^T + \lambda \boldsymbol{I}_n)^{-1}\boldsymbol{y}
\end{aligned}
\tag{20}
$$

where $\boldsymbol{I}_{p+\ell}$ is an $(p+\ell) \times (p+\ell)$-identity matrix and $\boldsymbol{X}\boldsymbol{X}^T$ is the $n \times n$ matrix.

Note that

$$
\boldsymbol{X}\boldsymbol{X}^T = \sum_{i=1}^{p} \boldsymbol{x}_i\boldsymbol{x}_i^T + \boldsymbol{H}_s\boldsymbol{H}_s^T = \boldsymbol{X}_{(-s)}\boldsymbol{X}_{(-s)}^T + \boldsymbol{H}_s\boldsymbol{H}_s^T, \quad \boldsymbol{H}_s = (h_{ij}) \in \Re^{n \times \ell}.
\tag{21}
$$

Let $\boldsymbol{A} = \boldsymbol{X}_{(-s)}\boldsymbol{X}_{(-s)}^T + \lambda \boldsymbol{I}_n$. Then applying the Sherman-woodbury inversion formula, we have

$$
\begin{aligned}
(\boldsymbol{X}\boldsymbol{X}^T + \lambda \boldsymbol{I}_n)^{-1} &= (\boldsymbol{X}_{(-s)}\boldsymbol{X}_{(-s)}^T + \lambda \boldsymbol{I}_n + \boldsymbol{H}_s\boldsymbol{H}_s^T)^{-1} = (\boldsymbol{A} + \boldsymbol{H}_s\boldsymbol{H}_s^T)^{-1} \\
&= \boldsymbol{A}^{-1} - \boldsymbol{A}^{-1}\boldsymbol{H}_s(\boldsymbol{I}_\ell + \boldsymbol{H}_s^T\boldsymbol{A}^{-1}\boldsymbol{H}_s)^{-1}\boldsymbol{H}_s^T\boldsymbol{A}^{-1}
\end{aligned}
\tag{22}
$$

Using the svd of $\boldsymbol{X}_{(-s)} = U\Sigma V^T$ where $U \in \Re^{n \times n}$ and $V \in \Re^{p \times p}$ are orothogonal matrices and $\Sigma \in \Re^{n \times p}$ are diagonal matrix with diagonal entries $\sigma_1 \geq \dots \geq \sigma_p \geq \sigma_{p+1} = \cdots = \sigma_n = 0$, we have

$$
\boldsymbol{A}^{-1} = (\boldsymbol{X}_{(-s)}\boldsymbol{X}_{(-s)}^T + \lambda \boldsymbol{I}_n)^{-1} = (U\Sigma\Sigma^T U^T + \lambda \boldsymbol{I}_n)^{-1} = U(\Sigma\Sigma^T + \lambda \boldsymbol{I}_n)^{-1}U^T
\tag{23}
$$

and

$$
\begin{aligned}
\boldsymbol{\xi} &= \boldsymbol{H}_s^T\boldsymbol{A}^{-1}\boldsymbol{H}_s = \boldsymbol{H}_s^T U(\Sigma\Sigma^T + \lambda \boldsymbol{I}_n)^{-1}U^T\boldsymbol{H}_s = \sum_{j=1}^{n} \frac{1}{\sigma_j^2 + \lambda}\boldsymbol{H}_s^T\boldsymbol{u}_j\boldsymbol{u}_j^T\boldsymbol{H}_s \\
\boldsymbol{\eta} &= \boldsymbol{H}_s^T\boldsymbol{A}^{-1}\boldsymbol{y} = \boldsymbol{H}_s^T U(\Sigma\Sigma^T + \lambda \boldsymbol{I}_n)^{-1}U^T\boldsymbol{y} = \sum_{j=1}^{n} \frac{1}{\sigma_j^2 + \lambda}\boldsymbol{H}_s^T\boldsymbol{u}_j\boldsymbol{u}_j^T\boldsymbol{y}.
\end{aligned}
\tag{24}
$$

Then the ridge estimate is

$$
\begin{aligned}
\hat{\boldsymbol{f}} &= \boldsymbol{X}\boldsymbol{X}^T(\boldsymbol{X}\boldsymbol{X}^T + \lambda \boldsymbol{I}_n)^{-1}\boldsymbol{y} = (\boldsymbol{X}\boldsymbol{X}^T + \lambda \boldsymbol{I}_n - \lambda \boldsymbol{I}_n)(\boldsymbol{X}\boldsymbol{X}^T + \lambda \boldsymbol{I}_n)^{-1}\boldsymbol{y} \\
&= \boldsymbol{y} - \lambda(\boldsymbol{X}\boldsymbol{X}^T + \lambda \boldsymbol{I}_n)^{-1}\boldsymbol{y} = \boldsymbol{y} - \lambda\boldsymbol{A}^{-1}\boldsymbol{y} + \lambda\boldsymbol{A}^{-1}\boldsymbol{H}_s(\boldsymbol{I}_\ell + \boldsymbol{H}_s^T\boldsymbol{A}^{-1}\boldsymbol{H}_s)^{-1}\boldsymbol{H}_s^T\boldsymbol{A}^{-1}\boldsymbol{y} \\
&= \boldsymbol{y} - \lambda\boldsymbol{A}^{-1}\boldsymbol{y} + \lambda\boldsymbol{A}^{-1}\boldsymbol{H}_s(\boldsymbol{I}_\ell + \boldsymbol{\xi})^{-1}\boldsymbol{\eta} \\
&= \boldsymbol{y} - \lambda U(\Sigma\Sigma^T + \lambda \boldsymbol{I}_n)^{-1}U^T\boldsymbol{y} + \lambda U(\Sigma\Sigma^T + \lambda \boldsymbol{I}_n)^{-1}U^T\boldsymbol{H}_s(\boldsymbol{I}_\ell + \boldsymbol{\xi})^{-1}\boldsymbol{\eta} \\
&= \boldsymbol{y} - \sum_{j=1}^{n} \frac{\lambda\boldsymbol{u}_j(\boldsymbol{u}_j^T\boldsymbol{y})}{\sigma_j^2 + \lambda} + \lambda \left( \sum_{j=1}^{n} \frac{\boldsymbol{u}_j\boldsymbol{u}_j^T\boldsymbol{H}_s}{\sigma_j^2 + \lambda} \right)(\boldsymbol{I}_\ell + \boldsymbol{\xi})^{-1}\boldsymbol{\eta}
\end{aligned}
\tag{25}
$$

where the $\boldsymbol{u}_i$ are the columns of $U \in \Re^{n \times n}$ and $\sum_{j=1}^{n} \boldsymbol{u}_j\boldsymbol{u}_j^T = \boldsymbol{I}_n$.

### 2.2.3 Fast Ridge estimate

The above computation involves $n$ sumations. To simplify the computations, we notice that $U = [U_1, U_2] \in \Re^{n \times n}$ where $U_1 = [\boldsymbol{u}_1, ..., \boldsymbol{u}_p] \in \Re^{n \times p}$ and $U_2 = [\boldsymbol{u}_{p+1}, ..., \boldsymbol{u}_n] \in \Re^{n \times (n-p)}$ and $U_1 \perp U_2$. Using the reduced SVD and $\sigma_{p+1} = \cdots = \sigma_n = 0$, we have the freedom to choose $U_2$ as long as $U_1 \perp U_2$ in such a way that the orthogonalization process is applied to the order of $\boldsymbol{h}_{:,k}$, $k = 1, ..., \ell$.

- $\boldsymbol{u}_j$ are orthogonal to $\boldsymbol{h}_{:,k}$, for all $j = p + k + 1, ..., n$
- $\boldsymbol{u}_{p+k}$ are the complements of the orthogonal projections of $\boldsymbol{h}_{:,k}$, onto $[U_1, \boldsymbol{u}_{p+k-1}]$, $k = 1, ..., \ell$, respectively and orthogonal to each other.

That is, for $k = 1, ..., \ell$,

$$\hat{\boldsymbol{u}}_{p+k} = (\boldsymbol{I}_n - P_{[U_1, \boldsymbol{u}_{p+k-1}]})\boldsymbol{h}_{:,k} = (\boldsymbol{I}_n - \sum_{i=1}^{p+k-1} \boldsymbol{u}_i \boldsymbol{u}_i^T)\boldsymbol{h}_{:,k} = \boldsymbol{h}_{:,k} - \sum_{i=1}^{p+k-1} \boldsymbol{u}_i(\boldsymbol{u}_i^T \boldsymbol{h}_{:,k})$$

$$\boldsymbol{u}_{p+k} = \hat{\boldsymbol{u}}_{p+k} / \|\hat{\boldsymbol{u}}_{p+k}\|. \tag{26}$$

Then $\boldsymbol{\xi}$ and $\boldsymbol{\eta}$ can be simplified as

$$\boldsymbol{\eta} = \sum_{j=1}^{p+\ell} \frac{1}{\sigma_j^2 + \lambda} \boldsymbol{H}_s^T \boldsymbol{u}_j \boldsymbol{u}_j^T \boldsymbol{y}$$

$$\boldsymbol{\xi} = \sum_{j=1}^{p+\ell} \frac{1}{\sigma_j^2 + \lambda} \boldsymbol{H}_s^T \boldsymbol{u}_j \boldsymbol{u}_j^T \boldsymbol{H}_s. \tag{27}$$

Therefore the ridge estimate can be further simplified as

$$\hat{\boldsymbol{f}} = \sum_{i=1}^{p} \frac{\sigma_i^2}{\sigma_i^2 + \lambda} \boldsymbol{u}_i(\boldsymbol{u}_i^T \boldsymbol{y}) + \lambda \left( \sum_{j=1}^{p+\ell} \frac{\boldsymbol{u}_j \boldsymbol{u}_j^T \boldsymbol{H}_s}{\sigma_j^2 + \lambda} \right) (\boldsymbol{I}_\ell + \boldsymbol{\xi})^{-1} \boldsymbol{\eta}. \tag{28}$$

## 3 Details on Scheme and Algorithm

CKKS scheme supports computation of approximate numbers by considering the noise for the hardness assumption as part of error that occurs during arithmetic. As a result, CKKS solves floating-point operations efficiently by introducing a bounded loss of precision. Considering that there is always a numerical error in the computation of a machine, this trade-off is beneficial for the purpose of machine learning. CKKS provides addition and multiplication operations between a ciphertext and a ciphertext, and between a ciphertext and plaintext. After each multiplication, rescaling is neededto manage the magnitude of the error. Since the rescaling reduces the ciphertext modulus, the number of operation is limited without bootstrapping. Also, CKKS supports SIMD operations by encoding a complex vector (with size at most $N/2$) into a ring element. It enables parallel operations, and with slot-wise rotation operation, the sum of the values in the different slots can be evaluated efficiently. However, since bigger slot size increases the bootstrapping time, the size of the ciphertext should be carefully selected. For detailed information of CKKS and its basic operations, we refer to Cheon et al. [2017].

Using the basic algorithms, InnerProduct and MatVecProduct in our main paper can be evaluated as Algorithm 1. In the algorithm, the notation of basic operation follows that of Park et al. [2020].

## 4 Extension to nonlinear models

Our effective Ridge regression method can be extended to nonlinear regression by considering linear combination of fixed nonlinear basis functions of the input variables. We omit a detailed

---

**Algorithm 1** Algorithms for inner product and matrix-vector multiplication

---

1: **procedure** InnerProduct($\boldsymbol{u}$, $c$)▷ calculate inner product of $\boldsymbol{u}$ and the decryption of ciphertext $c$
2: $\qquad c' \leftarrow \mathsf{CMult}(\boldsymbol{u}, \mathsf{c})$
3: $\qquad$ **for** $j$ in $1, \ldots, \log_2 n$ **do**
4: $\qquad\qquad tmp \leftarrow \mathsf{Rotate}(c', -2^j)$
5: $\qquad\qquad c' \leftarrow \mathsf{Add}(c', tmp)$
6: $\qquad$ **end for**
7: $\qquad$ **return** $c'$
8: **end procedure**
9: **procedure** MatVecProduct($\boldsymbol{U} = (\boldsymbol{u}_1 \cdots \boldsymbol{u}_p)$, $\boldsymbol{\sigma} = (\sigma_1, \cdots, \sigma_p)$, $\boldsymbol{V} = (\boldsymbol{v}_1 \ldots \boldsymbol{v}_p)$, $c$) $\qquad$ ▷
$\qquad$ calculate product of matrix $\boldsymbol{A} = \boldsymbol{U}\boldsymbol{\Sigma}\boldsymbol{V}^{\boldsymbol{T}}$ and the decryption of ciphertext $c$
10: $\qquad$ **for** $i$ in $1, \ldots, p$ **do**
11: $\qquad\qquad tmp \leftarrow \mathsf{InnerProduct}(\boldsymbol{v}_i, c)$
12: $\qquad\qquad tmp \leftarrow \mathsf{CMult}(\sigma_j \boldsymbol{u}_i, tmp)$
13: $\qquad\qquad$ **if** i=0 **then**
14: $\qquad\qquad\qquad c' \leftarrow tmp$
15: $\qquad\qquad$ **else**
16: $\qquad\qquad\qquad c' \leftarrow \mathsf{Add}(c', tmp)$
17: $\qquad\qquad$ **end if**
18: $\qquad$ **end for**
19: $\qquad$ **return** $c'$
20: **end procedure**

---

formulation, but in case of one private variable, by replacing $\boldsymbol{x}_i = (x_{i1}, \ldots, x_{ip}, x_{is})$ with $\phi(\boldsymbol{x}_i) = (\phi_1(\boldsymbol{x}_i), \ldots, \phi_m(\boldsymbol{x}_i), x_{is})$, the ridge estimate is

$$\hat{\boldsymbol{f}} = \boldsymbol{\Phi}(\boldsymbol{\Phi}^T\boldsymbol{\Phi} + \lambda\boldsymbol{I}_{M+1})^{-1}\boldsymbol{\Phi}^T\boldsymbol{y} = \boldsymbol{K}(\boldsymbol{K} + \lambda\boldsymbol{I}_n)^{-1}\boldsymbol{y} \tag{29}$$

where $\boldsymbol{\Phi} \in \mathbb{R}^{n \times (M+1)}$ whose i-th row is $\phi(\boldsymbol{x}_i)$. Defining the kernel function as

$$K(\boldsymbol{x}_i, \boldsymbol{x}_j) = \phi(\boldsymbol{x}_i)^T\phi(\boldsymbol{x}_j), \tag{30}$$

the kernel matrix can be easily obtained as

$$\boldsymbol{K} = \boldsymbol{\Phi}\boldsymbol{\Phi}^T = \sum_{i=1}^{p}\boldsymbol{\Phi}_i\boldsymbol{\Phi}_i^T + \boldsymbol{h}\boldsymbol{h}^T = \boldsymbol{\Phi}_{(-s)}\boldsymbol{\Phi}_{(-s)}^T + \boldsymbol{h}\boldsymbol{h}^T = \boldsymbol{K}_{(-s)} + \boldsymbol{h}\boldsymbol{h}^T \tag{31}$$

where $\boldsymbol{h}$ is defined the same as in section 2.1. Then all the remaining steps are the same as in section 2.1, with $\boldsymbol{X}\boldsymbol{X}^{\boldsymbol{T}}$ replaced by $\boldsymbol{K}$.

We validated the nonlinear method on the same datasets as in our main paper. To give nonlinearity to the teacher model, we trained a neural network which consists of three fully-connected layers as the teacher model. Each layer reduces the data dimension to 8, 4, 1, respectively, and a sigmoid function was taken after the last layer. After the first two layers, we used square activation instead of ReLU for an HE-friendly inference. The step 3 of our method was trained with the kernel Ridge regression demonstrated above. The results are summarized in Table 1. It is shown that the nonlinear extension works well with every dataset, achieving at least as high accuracy as our original method. This is because linear Ridge regression cannot represent the nonlinearity obtained from the nonlinear teacher model well. In the case of **Cancer**, it seems that the linear boundary is close to the optimum, so the same performance is obtained even if nonlinearity is not introduced. **Ours-nonlinear** will perform better if we use a more complex teacher model or a dataset where the linear teacher model performs poorly.

## References

J. H. Cheon, A. Kim, M. Kim, and Y. Song. Homomorphic encryption for arithmetic of approximate numbers. In *International Conference on the Theory and Application of Cryptology and Information Security*, pages 409–437. Springer, 2017.

Table 1: Results for our nonlinear regression model

| Dataset | Accuracy (%) | | | |
|---|---|---|---|---|
| | NN (step 1) | LRHE | Ours-linear | Ours-nonlinear |
| **Adult** | 83.90 | 80.106 | 81.217 | **83.803** |
| **Bank** | 87.72 | 87.058 | 87.279 | **87.943** |
| **Cancer** | 94.85 | 94.118 | **94.853** | **94.853** |
| **Diabetes** | 75.68 | 70.130 | 75.325 | **76.623** |
| **Credit** | 88.41 | 86.232 | 86.232 | **89.623** |

M. Mohri, A. Rostamizadeh, and A. Talwalkar. *Foundations of machine learning*. MIT press, 2018.

S. Park, J. Byun, J. Lee, J. H. Cheon, and J. Lee. He-friendly algorithm for privacy-preserving svm training. *IEEE Access*, 8:57414–57425, 2020.