# OpenReview forum: "Parameter-free HE-friendly Logistic Regression"
_NeurIPS.cc/2021/Conference — NeurIPS 2021 Poster_

### Official Review · Reviewer_WfvM · 2021-07-11

**Rating:** 6
**Confidence:** 3

**Summary:**

This paper tackles the problem of performing logistic regression in a homomorphic encryption setting. The propose the following:

- Pretrain a regular logistic regression on some public data (completely plaintext)

- Run this network on the private data to produce soft targets (and shift the means of the soft targets to deal with datashift shift, with a hyper-parameter)

- Solve ridge regression (possibly with some non-private variables) to estimate these targets on the private data

The authors provide a fast solution for step 3 and show that their model can outperform LRHE in accuracy and speed on different datasets from the UCI repository.


**Limitations And Societal Impact:**

The paper notes that this framework cannot be used for nonlinear models easily (like NNs), however, it can be used with kernel methods.  The study does not note possible negative social impacts.

**Main Review:**

Originality:
The authors propose to tackle the problem of hyper-parameter selection in homomorphic encryption by leveraging a plaintext-text dataset at the start of the training process. This offers an interesting novel direction to explore.

Quality / Significance:

The core claims of this paper are:

*Introducing a hyper-parameter free version of HE logistic regression

However, the paper introduces a hyper-parameter lamba for their ridge regression which still needs to be selected somehow. Perhaps I’m misunderstanding something but there seems to be a hyper-parameter here.

*This method is faster and more accurate than prior work

I have a few questions here. Is the LRHE baseline also adapted to only use a few private variables? It seems that the number of private variables here is critical to runtime and so a head to head comparison where all variables are private is critical.  Moreover, how does LRHE use the plaintext data? Does it not use it all? If so, that isn't a fair comparison to the model here.

Also, how does it compare to other mechanisms for using the plaintext data? For example, can the pretrained teacher be used as an initial guess of weights for both LRHE and this approach? Moreover, how does it perform by itself without any training on the private data (in table 1)?

As of now, it is difficult to assess the significance of these results.


**Time Spent Reviewing:**

2

---

> ### Author Response · Authors · 2021-08-10
> **Response to reviewer WfvM**
>
> $\textbf{Reponse}$
>
> We would like to thank the reviewers for their input. Their comments have been thoroughly considered, and by altering the manuscript in accordance with these comments, we are convinced that the quality of our paper will be significantly improved.
>
> **C1 : The paper introduces a hyper-parameter lamba for their ridge regression which still needs to be selected somehow.**
>
> A1 : Thank you for the comment. The main argument in our paper is that hyperparameter tuning using HE is very inefficient, and we proposed a method to avoid tuning the learning rate and sigmoid approximation range, parameters that have a critical effect on the logistic regression training performance. Although lambda is also a hyperparameter, It is well known that values of lambda ​​within a certain range do not significantly affect the performance. Therefore, the value of lambda is often fixed in ML studies. Our reason to introduce lambda in our formulation is to guarantee the non-singularity of matrix A in equation (6). In our experiments, we got better accuracy performance compared to LRHE even though we used lambda=1 without tuning. Also, we verified with the datasets used in paper that changing the value of lambda in {0.1, 1, 10, 100} does not change the accuracy of our method at all for 4 datasets except the Cancer dataset. For the Cancer dataset, only setting lambda=100 resulted in about a 4%p of accuracy drop.
> Nevertheless, the reviewer's point that the expression "parameter-free" may confuse readers is correct, and we will replace it with a relaxed expression to eliminate confusion.
>
> **C2 : Is the LRHE baseline also adapted to only use a few private variables?**
>
> A2 : Thank you for the comment. As demonstrated in 105-108 of our main paper, LRHE baseline requires the weighted sum of all variables as an input for sigmoid function. Even though encrypting only private variables with HE, the weighted sum of plaintext and ciphertext results in a ciphertext. Therefore, LRHE does not benefit much from partial encryption of private variables. Modifying LRHE to make it suitable for partial encryption is another topic of research, and we hope the reviewer agrees that it is beyond our scope.
> In fact, in the experiment, for LRHE we encrypted all the private variables in the same way as Ours-grad, which is a modified version of our method to encrypt all the variables. In response to the reviewer’s comments, we additionally implemented LRHE with only private variables encrypted, and verified that the computation time decreases less than 1.2% for all datasets. The results will be added in the revised version.
>
> **C3 : How does LRHE use the plaintext data?**
>
> A3 : Thank you for the suggestion. We consider that it is our novel contribution to use the model trained using plaintext for the encrypted model. Therefore, it is difficult for other models including LRHE to benefit from plaintext data. Following the reviewer's suggestion, we experimented using the pretrained teacher as the initial guess of the weight, but there was no change in the performance of the LRHE. This is because the loss function for logistic regression is convex and the training converges well with appropriate learning rate. The reason that our method shows better accuracy than LRHE is the approximation of sigmoid function in LRHE, not that LRHE doesn’t converge well.
>
> **C4 : How does it perform by itself without any training on the private data (in table 1)?**
>
> A4 : Thank you for the suggestion. If our understanding is correct, the inference performance of the teacher model on encrypted data is shown in Table 2 (column 2 - LR with X1). It is shown that when the underlying distribution of plaintext data and encrypted data are different, our method performs significantly better than the teacher model for encrypted data.
>
> **C5 : The study does not note possible negative social impacts.**
>
> A5 : Thank you for the comment. This paper is a study on how to extend the existing machine learning method in the privacy-preserving direction, so we think that it essentially has a positive societal impact. However, as reviewer XQWo pointed out, our security model is limited to semi-honest, non colluding participants and thus our model can be corrupted by malicious participants and may have a negative social impact. We will add some discussion about this point in the revised version.
>
> **Additional response for reviewer WfvM**
>
> We feel sorry that the core argument of our paper may not have been sufficiently conveyed to the reviewer. The methods we used (efficient algorithm for partial encryption or utilizing published teacher model for encrypted training) have not been proposed before, and therefore, a baseline that can be compared under the exact same conditions may not yet exist. Nevertheless, we tried to proceed the experiment in a direction that is not disadvantageous to LRHE baseline as much as possible. For example, for LRHE we cheated to find the learning rate and sigmoid approximation range without encryption. If LRHE was performed several times for parameter validation, the computation time would be much longer.
> We hope that the reviewer reconsiders the evaluation of the paper according to the responses.

---

### Official Review · Reviewer_WSkd · 2021-07-15

**Rating:** 5
**Confidence:** 3

**Summary:**

This study considers a situation that a part of unencrypted samples are available while the other samples are provided with encrypted by FHE (or LHE). First, the modeler trains a logistic regression model with unencrypted samples. By using this model, given encrypted samples,  the modeler obtains the encrypted logits. To absorb the difference of sample generating distributions of non-private samples and private samples, moment matching is applied on encrypted logits (I could not understand why this works).  Then, ridge regression is solved with the encrypted logits with labels, in that the solution can be obtained in an analytic form instead of iterative methods as logistic regression regularly requires.
**Due to the lack of a detailed description of the security model and global information flow of the protocol, this summary might be wrong.


**Ethical Concerns:**

No.

**Limitations And Societal Impact:**

No.
I think this type of study is originally presented to enhance privacy, no need to support it.


**Main Review:**

In my understanding, this paper does not contain a technically original contribution, while a practical solution for logistic regression model training over FHE is presented by combining known techniques in matrix algebra and cryptography. In this sense, this paper presented a meaningful construction of logistic regression over FHE.

The weak point of this manuscript is that the contents of the paper overly emphasize the detailed mathematical operations while necessary information to understand the security model considered in this study is lost. Due to the lack of this, the reviewer could not understand the information flow between data owner, modeler, and crypto-service provider. Also, the reviewer could not understand who wanted to protect what from whom by this protocol. For this reason, the reviewer thinks it is not appropriate this publish this manuscript in its current form.

Line 130
...we should assume that M should possess unencrypted D1 that can be used for step 1

This paper does not give a specific security setting (who does what and what is private). Without a clear description of the security model, it is impossible to see if the presented scheme is secure or not.


Also, line 120
... The details of our protocol to achieve the security goal is in Appendix.

The protocol itself is the main contribution of this work so it should be contained in the main body.

Line142
... we suggest adding a regularization term β on l2 so that it can take into account the difference between two distributions

This does not make sense. Why regularization term can resolve the difference between two distributions?

Line 148
...By encrypting a small private portion of the whole information,

Does this mean that some portion (samples? variables?) is private and the other (samples? variables?) are not private?  Who decides which is private and which is not?


**Time Spent Reviewing:**

3

---

> ### Author Response · Authors · 2021-08-10
> **Response to reviewer WSkd**
>
> $\textbf{Reponse}$
>
> We would like to thank the reviewers for their input. Their comments have been thoroughly considered, and by altering the manuscript in accordance with these comments, we are convinced that the quality of our paper will be significantly improved.
>
> **C1 : This paper does not contain a technically original contribution.**
>
> A1 : As the reviewer mentioned, the main contribution of our paper is a practical solution for training HE-friendly logistic regression which avoids HE-impractical parameter selection.
> Our study can also be regarded as a proposal for a novel knowledge distillation technique for a linear model. From this point of view, we consider that there are two technical contributions to our research:
>
> 1. We showed that the information from plaintext can be distilled to encrypted data to enhance the accuracy and efficiency of HE-based training.
> 2. By using linear methods, we provided the generalization bound (Theorem 2) of our method, which is one of the biggest advantages of the linear methods compared to deep learning. In particular, the $\mathcal{H}$-divergence term in the theorem implies that we should reduce the gap between two distributions to achieve better performance, and we proposed a simple mean matching method to deal with it.
>
> Recently there has been massive research on HE-based machine learning from logistic regression [4] to deep learning [5-8], but as far as we know, none of them offered any theoretical claims other than computational tricks. Compared to others, our study has an advantage in that the proposed method can be evaluated theoretically.
>
> **C2 : Necessary information to understand the security model considered in this study is lost.**
>
> A2 : We deeply agree with the reviewer's opinion that our security model and the protocol to achieve it have not been sufficiently covered in the paper. Taking the reviewer's advice, we will reduce the mathematical description in the main paper and explain our protocol sufficiently.
> To help the reviewer understand, we briefly demonstrate our protocol here. It consists of four steps:
>
> 1. (Teacher modeling) modeler $\mathsf{M}$ trains a teacher model $f\_s$ with unencrypted dataset $\mathcal{D}\_1$, where $\mathcal{D}\_1$  is considered to be owned by $\mathsf{M}$ or publicly available.
>
> 2. (Encryption) crypto service provider $\mathsf{C}$ generates keys $(pk, sk)$ and sends $pk$ to $\mathsf{O}$ and $\mathsf{M}$. $\mathsf{O}$ encrypts private variables of their dataset $\mathcal{D}\_2$ and send it to $\mathsf{M}$.
>
> 3. (Training on encrypted data) $\mathsf{M}$ infers encrypted logit $\mathsf{Enc}(l_2) = f_s(\mathcal{D}\_2)$ and by mean matching evaluates $\mathsf{Enc}(\tilde{l\_2}) =\mathsf{Enc}( l\_2)+\mathsf{Enc}(\beta)$. $\mathsf{M}$ trains privacy-preserving ridge regression on $\mathcal{D}\_2$ and $\mathsf{Enc}(\tilde{l\_2})$ and obtains $\mathsf{Enc}(\omega)$.
>
> 4. (Decryption) $\mathsf{M}$ generates a random polynomial $r$ and send $\mathsf{Enc}(\omega+r)$ to $\mathsf{C}$. $\mathsf{C}$ decrypts $w+r$, and add a random discrete Gaussian noise $e$ and send $w+r+e$ back to $\mathsf{M}$. $\mathsf{M}$ subtracts $r$ and the final weight is obtained as $w+e$.
>
> In the protocol $r$ protects $\omega$ against $\mathsf{C}$. $e$ is added to defend against attack proposed in [1], and according to [2] with a high probability it causes $<1$ bits of precision loss.
> In addition, we will provide the security proof which guarantees the security of our protocol under the assumption that $\mathsf{M}$ and $\mathsf{C}$ does not collude with each other and the underlying HE scheme is semantically secure against passive adversary.
> Also, we will cover the private inference phase in the revised version. To prevent the data owner from getting information about the model from the inference result, $\mathsf{M}$ can generate a random positive $r’$ from the plaintext domain and multiplies it to the inference result.
>
> Our answer for other comments are :
> The “portion” means variables, not samples, in our study. Of course, research to protect the privacy of partial samples will also be valuable.
> In our protocol, the data owners $\mathsf{O}$ can decide which variables are private based on their own criterion. As we mentioned in Chapter 2 of our main paper, the criterion varies according to research area.
>
>
> **C3 : Why regularization term can resolve the difference between two distributions?**
>
> A3 : Thank you for your comment. We agree with the reviewer that we did not provide enough explanation about our mean matching. As we mentioned above, our idea of mean matching started from our theorem. The second term in Theorem 2 indicates that we should reduce the distribution gap between the plaintext data and encrypted data. To reduce $\mathcal{H}$-divergence, we applied a simplified version of kernel mean matching from [3], which is widely used in domain adaptation studies, to the logits.
>
> The original kernel mean matching tries to match the means of the distributions of X in some kernel space. Thus, our mean matching can be seen as applying a simplified version that does not consider kernel space. However, we found out that there is a problem when directly applying mean matching to logits - if the distributions of logits become similar, the resulting distributions of predictions also become similar, which has a negative effect on moving the prediction away from the true label. Therefore, we multiplied the logit by a weight that can reflect the distribution of the true label. Since the true label has a binary value, summation can reflect the distribution in a simple but effective way. Through Figure 3, we have shown the effectiveness of mean matching on empirical experiments.
>
> $\textbf{References}$
>
> [1] Li, B., & Micciancio, D. (2021, October). On the security of homomorphic encryption on approximate numbers. In Annual International Conference on the Theory and Applications of Cryptographic Techniques (pp. 648-677). Springer, Cham.
>
> [2] Cheon, J. H., Hong, S., & Kim, D. (2020). Remark on the Security of CKKS Scheme in Practice. IACR Cryptol. ePrint Arch., 2020, 1581.
>
> [3] Gretton, A. (2009). Kernel approaches to covariate shift.
>
> [4] Kim, A., Song, Y., Kim, M., Lee, K., & Cheon, J. H. (2018). Logistic regression model training based on the approximate homomorphic encryption. BMC medical genomics, 11(4), 23-31.
>
> [5] Lou, Q., & Jiang, L. (2019). SHE: A Fast and Accurate Deep Neural Network for Encrypted Data. Advances in Neural Information Processing Systems, 32, 10035-10043.
>
> [6] Lou, Q., Bian, S., & Jiang, L. (2020). AutoPrivacy: Automated Layer-wise Parameter Selection for Secure Neural Network Inference. Advances in Neural Information Processing Systems, 33.
>
> [7] Lou, Q., Feng, B., Charles Fox, G., & Jiang, L. (2020). Glyph: Fast and Accurately Training Deep Neural Networks on Encrypted Data. Advances in Neural Information Processing Systems, 33.
>
> [8] Lou, Q., Lu, W. J., Hong, C., & Jiang, L. (2020). Falcon: Fast Spectral Inference on Encrypted Data. Advances in Neural Information Processing Systems, 2364-2374.

---

> ### Author Response · Authors · 2021-08-31
> **We are looking forward to your response**
>
> Dear Reviewer WSkd:
>
> Your feedback is highly appreciated and will help us to improve our work. Please let us know if our response is not clear enough to answer your questions, or if there are any more points that we need to explain. We are looking forward to your response.

---

### Official Review · Reviewer_XQWo · 2021-07-16

**Rating:** 7
**Confidence:** 3

**Summary:**

This paper introduces an interesting new approach for learning a classifier on homomorphically encrypted data. A few important assumptions are made: only very few features (one to three maybe) are privacy-protected and the modeler is able to build a regression model on a part of the dataset. Next, further training data including private features are used to convert the initial model into a Ridge regression model. The key innovation is that the encrypted computation step can be isolated to make it very simple when few features are encrypted.

**Limitations And Societal Impact:**

Training on encrypted data can raise all kinds of new concerns. For example, poisoning becomes really challenging to prevent, as input data can't be validated by the modeler. This may or may not matter, depending on the exact use-case. Maybe some paragraph discussing concrete scenarios, where the proposed system makes sense, could be added.

**Main Review:**

I think the idea is very interesting and hits many of the pain-points in learning on encrypted data. Realistically not all features are equally private, so it's great to see that this can be taken advantage of so well. Another huge advantage is getting rid of the sigmoid approximation, which is always problematic due to inability to control the input range.

This work seems totally novel and interesting to me, but I'm not enough of an expert to evaluate whether (from an ML point-of-view) the approach that is taken is actually sound. The encrypted computing part is rather simple and could be done with a number of techniques to achieve semi-honest security.

**Time Spent Reviewing:**

3

---

> ### Author Response · Authors · 2021-08-10
> **Response to reviewer XQWo**
>
> $\textbf{Reponse}$
>
> We would like to thank the reviewers for their input. Their comments have been thoroughly considered, and by altering the manuscript in accordance with these comments, we are convinced that the quality of our paper will be significantly improved.
>
> C1 : The encrypted computing part is rather simple and could be done with a number of techniques to achieve semi-honest security.
>
> A1 : We thank the reviewer for appreciating the novelty of our approach. As reviewer mentioned, we considered several techniques to achieve semi-honest security for our protocol.
>
> 1. After modeler $\mathsf{M}$ trains Ridge regression and obtains encrypted weight $\mathsf{Enc}(\omega)$, $\mathsf{M}$ adds a random polynomial $r$ to keep security of $\mathsf{Enc}(\omega)$ against crypto service provider $\mathsf{C}$.
>
> 2. After $\mathsf{C}$ decrypts $\mathsf{Enc}(\omega+r)$ $\mathsf{C}$ adds a discrete Gaussian noise e and send $\omega+r+e$ back to $\mathsf{M}$. Therefore the final weight used for inference is $\omega+e$. $e$ prevents attack against CKKS cryptosystem proposed in [1].
>
> In the revised version of our paper we will demonstrate our protocol in more detail along with a security proof of our protocol.
>
> C2 : For example, poisoning becomes really challenging to prevent, as input data can't be validated by the modeler.
>
> A2 : Thank you for the claim. As reviewer pointed out, there is a possibility of corrupted data owners $\mathsf{O}$ who send malicious encrypted data that adversely affects the training. As far as we know, those possibility has not been covered in the study on data outsourcing using homomorphic encryption. The assumption of our protocol is that $\mathsf{M}$ and $\mathsf{C}$ are honest-but-curious and do not collude with each other, and it is thought that an additional assumption about $\mathsf{O}$'s honesty will prevent such attacks. We agree with the reviewer that the assumption holds depending on the use-case. Specifically, it would not be very risky to make this assumption if the data owner would benefit from the trained model. We will add this discussion to our paper.
>
> $\textbf{References}$
>
> [1] Li, B., & Micciancio, D. (2021, October). On the security of homomorphic encryption on approximate numbers. In Annual International Conference on the Theory and Applications of Cryptographic Techniques (pp. 648-677). Springer, Cham.

---

> ### Author Response · Authors · 2021-08-31
> **We are looking forward to your response**
>
> Dear Reviewer XQWo:
>
> Your feedback is highly appreciated and will help us to improve our work. Please let us know if our response is not clear enough to answer your questions, or if there are any more points that we need to explain. We are looking forward to your response.

---

### Official Review · Reviewer_xo2J · 2021-07-26

**Rating:** 6
**Confidence:** 3

**Summary:**

The paper tackles with the hyper-parameter tuning problem in logistic regression on encrypted data. Assuming a fraction of public data and a fraction of public features, the authors show by experiments that similar or better accuracy can be achieved with ~3x speed.

**Limitations And Societal Impact:**

The authors have adequately addressed the limitations and potential negative societal impact of their work

**Main Review:**

I find the classification-to-regression idea and the computation trick quite interesting. The speed-up is significant and evident. I think this part is solid work and makes a good publication. However, I have to say that as a non-expert, I cannot judge the novelty of the idea and the trick. In particular, I feel that this classification-to-regression idea may generally be useful in domain adaptation / transfer learning. I cannot tell if it already appeared in that community. Secondly, I suppose the linear algebraic trick used in lines 150-176 was at least partially known. For example, I suppose Equation (4) should be well-known, but is it so well-known that no reference is needed/possible? Some notes would definitely help readers like me.

On the other hand, I have one major criticism: The theory in section 4 doesn't seem to make any contribution to the paper. It also doesn't make much sense to me. I recommend removing this part, together with the claim that the authors find a new bound. Otherwise, section 4 needs a significant improvement at the very least. For example, in Theorem 2, I don't know what is $\\tilde\{H\}\_0$. The definition in line 214 doesn't make much sense to me. According to line 209, $R_{D_T}$ relies on a ground truth $c_T$. What is it? Why are we interested in the divergence defined in line 214? How does $H$ and $\tilde{H}$ effect the divergence? Is it an original definition or is there a source? I couldn't find it in the book cited. In the proof of Theorem 2, the authors cited Theorem 13.2 of the book, but that deals with multi-class case. Isn't 13.4 a better fit for the binary problem in the paper? I'd like to see the response on this issue and am open to change my rating.

Finally, a comment on the presentation as a non-expert reader: It takes a bit effort for me to figure out who does what and who sees what and why the protocol is secure. Stating these facts with explicit reference to Figure 2 and/or Algorithm 1 may help.



**Time Spent Reviewing:**

6

---

> ### Author Response · Authors · 2021-08-10
> **Response to reviewer xo2J**
>
> $\textbf{Response}$
>
> We would like to thank the reviewers for their input. Their comments have been thoroughly considered, and by altering the manuscript in accordance with these comments, we are convinced that the quality of our paper will be significantly improved.
>
>
> **C1 : Is classification-to-regression idea generally useful in domain adaptation / transfer learning?**
>
> A1 : We used the classification-to-regression idea in this paper because we found an efficient regression technique for encrypting private variables with HE, which cannot be directly applied to classification. By introducing the classification-to-regression idea we could change the classification task by solving the closed form of regression problems. However, in traditional domain adaptation or transfer learning tasks, there is no need to change the classification task to regression since the gradient descent approaches are possible in plaintext problems. Therefore, even though the overall framework of our proposed method might look similar to traditional transfer learning, it is a novel method that is suitable for the HE tasks.
>
> **C2 : Is the linear algebraic trick used in the paper so well-known that no reference is needed/possible?**
>
> A2 : As we know, the linear algebraic tricks used in our paper including singular value decomposition and Sherman-Woodbury-Morrison formula are very well-known in ML society. We don't think this part needs a reference, but for the unfamiliar readers we'll add a more detailed explanation in the Appendix.
>
> **C3 :  What is $\tilde{\mathcal{H}}\_0$ and $c_T$?**
>
> A3 : Thank you for your remark. We found out that the unfamiliar term $\tilde{\mathcal{H}}\_0$ and the ground truth $c_T$ are not critical for our proof, and since they can confuse the readers, we modified the generalization upper bound without them. We have newly defined the risk of distribution $D\_{\mathcal{T}}$ as $\mathcal{R}\_{\mathcal{D}\_{\mathcal{S}}} = \mathbb{E}\_{\mathbf{z} \sim D\_{\mathcal{T}}} [1\_{\xi\_h(z)\leq0}]$. Then, previous $\tilde{\mathcal{H}}\_0$ can be substituted by the $\mathcal{H}$-divergence as in Ben-David et al. [1] as follows:
>
> $D\_{\mathcal{H}} (\mathcal{D}\_{\mathcal{S}},\mathcal{D}\_{\mathcal{T}}) = \sup\_{h,h' \in \mathcal{H}} |\mathcal{R}\_{\mathcal{D}\_{\mathcal{S}}}(h’) - \mathcal{R}\_{\mathcal{D}\_{\mathcal{S}}}(h) |$
>
> With this expression, we can modify the Theorem 1 and 2 by using the $\mathcal{H}$-divergence. We will change the manuscript in the revised version.
>
> **C4 : In the proof of Theorem 2, the authors cited Theorem 13.2 of the book, but that deals with multi-class case. Isn't 13.4 a better fit for the binary problem in the paper?**
>
> A4 : Thanks for the suggestion. Theorem 13.4 of [5] can be seen as a specific case of Theorem 13.2. Although our conclusion more closely resembles that of Theorem 13.4 we used both theorem’s claims in our proof. It is our mistake not to cite both theorems, and we will correct it in the revised version.
>
> **C5 : The theory in section 4 doesn't seem to make any contribution to the paper. It also doesn't make much sense to me.**
>
> A5 : Thank you for the comment. Knowledge distillation has been widely used in deep learning society to train efficient models while increasing performance, but our study suggested that it can have new meanings (information from unencrypted data can be used for training using encrypted data to enhance performance) when brought into relatively simple models. By using linear methods, our method is not only suitable for HE, but also can present a new generalization bound for distillation. Compared to that in [5], the generalization bound has been modified to fit into our method, so it has a contribution to our study.
>
> It is our proposed mean matching that serves as a stepping stone between our Theorem 2 and the actual method. In particular, the $\mathcal{H}$-divergence term in the theorem implies that we should reduce the gap between two distributions to achieve better performance. To reduce $\mathcal{H}$-divergence, we applied a simplified version of kernel mean matching from [2] to the logits, which resulted in an effective correction for the performance degradation caused by the distribution difference. We recognize that the contribution of the theorem has not been sufficiently explained in the main paper, and we will add a detailed explanation along with appropriate reference in the revised version.
>
> **C6 : It takes a bit effort for me to figure out who does what and who sees what and why the protocol is secure.**
>
> A6 : We feel deeply responsible for not communicating enough information about our protocol and security model to our readers. In the revised version we will state the protocol to keep security of our model in the main paper and revise Figure 2 to better describe the protocol.
>
> To explain for the reviewer, our protocol consists of four steps:
>
> 1. (Teacher modeling) modeler $\mathsf{M}$ trains a teacher model $f\_s$ with unencrypted dataset $\mathcal{D}\_1$, where $\mathcal{D}\_1$  is considered to be owned by $\mathsf{M}$ or publicly available.
>
> 2. (Encryption) crypto service provider $\mathsf{C}$ generates keys $(pk, sk)$ and sends $pk$ to $\mathsf{O}$ and $\mathsf{M}$. $\mathsf{O}$ encrypts private variables of their dataset $\mathcal{D}\_2$ and send it to $\mathsf{M}$.
>
> 3. (Training on encrypted data) $\mathsf{M}$ infers encrypted logit $\mathsf{Enc}(l_2) = f_s(\mathcal{D}\_2)$ and by mean matching evaluates $\mathsf{Enc}(\tilde{l\_2}) =\mathsf{Enc}( l\_2)+\mathsf{Enc}(\beta)$. $\mathsf{M}$ trains privacy-preserving ridge regression on $\mathcal{D}\_2$ and $\mathsf{Enc}(\tilde{l\_2})$ and obtains $\mathsf{Enc}(\omega)$.
>
> 4. (Decryption) $\mathsf{M}$ generates a random polynomial $r$ and send $\mathsf{Enc}(\omega+r)$ to $\mathsf{C}$. $\mathsf{C}$ decrypts $w+r$, and add a random discrete Gaussian noise $e$ and send $w+r+e$ back to $\mathsf{M}$. $\mathsf{M}$ subtracts $r$ and the final weight is obtained as $w+e$.
>
> In the protocol $r$ protects $\omega$ against $\mathsf{C}$. $e$ is added to defend against attack proposed in [3], and according to [4] with a high probability it causes $<1$ bits of precision loss.
> In addition, we will provide the security proof which guarantees the security of our protocol under the assumption that $\mathsf{M}$ and $\mathsf{C}$ does not collude with each other and the underlying HE scheme is semantically secure against passive adversary.
>
> $\textbf{References}$
>
> [1] Ben-David, S., Blitzer, J., Crammer, K., Kulesza, A., Pereira, F., & Vaughan, J. W. (2010). A theory of learning from different domains. Machine learning, 79(1), 151-175.
>
> [2] Gretton, A., Smola, A., Huang, J., Schmittfull, M., Borgwardt, K., & Schölkopf, B. (2009). Covariate shift by kernel mean matching. Dataset shift in machine learning, 3(4), 5.
>
> [3] Li, B., & Micciancio, D. (2021, October). On the security of homomorphic encryption on approximate numbers. In Annual International Conference on the Theory and Applications of Cryptographic Techniques (pp. 648-677). Springer, Cham.
>
> [4] Cheon, J. H., Hong, S., & Kim, D. (2020). Remark on the Security of CKKS Scheme in Practice. IACR Cryptol. ePrint Arch., 2020, 1581.
>
> [5] Mohri, M., Rostamizadeh, A., & Talwalkar, A. (2018). Foundations of machine learning. MIT press.

---

> ### Author Response · Authors · 2021-08-31
> **We are looking forward to your response**
>
> Dear Reviewer xo2J:
>
> Your feedback is highly appreciated and will help us to improve our work. Please let us know if our response is not clear enough to answer your questions, or if there are any more points that we need to explain. We are looking forward to your response.

---

### Decision · Program_Chairs · 2021-09-27

**Decision:**

Accept (Poster)

**Comment:**

This paper presents an interesting solution to the problem of training models on private data. The progress is made by first noticing that not all features are equally sensitive and then using it to reduce the classification problem, which is hard to train using HE to a regression problem that is easier to solve.
Most reviewers agreed that the paper makes a nice contribution to the field and should be accepted. The presentation of the work is fine but could be improved. Adding to comments made by the reviewers note, for example, that on line 122 “knowledge distillation” is discussed as if it was already presented before. The line reads “…which mimics the first phase of the knowledge distillation in that …” but this is the first-time knowledge distillation is being discussed. In this case, removing the “the” before “knowledge distillation” and adding a reference to a relevant paper could help the reader. We encourage the authors to revisit the paper and try to improve it readability.